# Uncertainty-Based Extensible Codebook for Discrete Federated Learning in Heterogeneous Data Silos

**Tianyi Zhang** [1]  **Yu Cao** [1]  **Dianbo Liu** [2]

## Abstract

Federated learning (FL), aimed at leveraging vast distributed datasets, confronts a crucial challenge: the heterogeneity of data across different silos. While previous studies have explored discrete representations to enhance model generalization across minor distributional shifts, these approaches often struggle to adapt to new data silos with significantly divergent distributions. In response, we have identified that models derived from FL exhibit markedly increased uncertainty when applied to data silos with unfamiliar distributions. Consequently, we propose an innovative yet straightforward iterative framework, termed *Uncertainty-Based Extensible-Codebook Federated Learning (UEFL)*. This framework dynamically maps latent features to trainable discrete vectors, assesses the uncertainty, and specifically extends the discretization dictionary or codebook for silos exhibiting high uncertainty. Our approach aims to simultaneously enhance accuracy and reduce uncertainty by explicitly addressing the diversity of data distributions, all while maintaining minimal computational overhead in environments characterized by heterogeneous data silos. Extensive experiments across multiple datasets demonstrate that UEFL outperforms state-of-the-art methods, achieving significant improvements in accuracy (by 3%–22.1%) and uncertainty reduction (by 38.83%–96.24%). The source code is available at https://github.com/destiny301/uefl.

## 1. Introduction

*Federated Learning (FL)*, well known for its capacity to harness data from diverse devices and locations—termed data

---
[1]University of Minnesota, MN, USA [2]National University of Singapore, Singapore. Correspondence to: Tianyi Zhang <zhan9167@umn.edu>, Dianbo Liu <dianbo@nus.edu.sg>.

*Proceedings of the 42nd International Conference on Machine Learning*, Vancouver, Canada. PMLR 267, 2025. Copyright 2025 by the author(s).

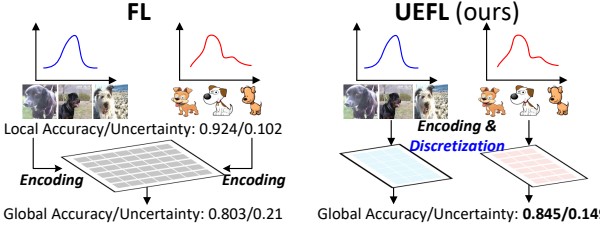

Figure 1: In the case of heterogeneous data silos, the global model of regular FL performs poorly compared to local models. By discretizing different domains into distinct latent spaces, our UEFL improves both accuracy and uncertainty. The reported values in the figure represent the average accuracy and uncertainty across the various data silos.

silos—while ensuring privacy, has become increasingly crucial in the digital era, particularly with the explosion of data from mobile sources. Despite its pivotal role in distributed computing, FL confronts a formidable challenge: the heterogeneity of data across different silos. Such diversity often results in a significant performance gap when integrating updates from local models into the global model. In Figure 1, we compare the mean accuracy of local FL models with that of the global model after integration when addressing data silos with different distributions. While local models may perform impressively within their own data domains, the aggregated global model often struggles to achieve similar performance levels after synthesizing updates from these varied data sources. This issue is especially pronounced in FL due to its reliance on varied data sources.

Recent studies (Ghosh et al., 2020; Agarwal et al., 2021; Liu et al., 2021; Kairouz et al., 2021a; Zhang et al., 2022; Yuan et al., 2022) have made significant advancements in addressing data heterogeneity within FL, with one notable approach being the use of discrete representations to enhance model robustness against minor data shifts. Nonetheless, this strategy struggles to generalize models to data silos exhibiting significant distributional differences. Furthermore, these methods face difficulties in adapting to unseen data distributions, as they typically require the entire model to be re-trained. Such constraints limit their flexibility in adapting to the dynamically changing data landscapes, posing

challenges for their applicability in real-world scenarios.

Moreover, we identify another critical issue impacting the model's performance across diverse data silos: increased uncertainty, as shown in Figure 1. The global model's accuracy not only deteriorates, but its uncertainty also trends upwards, signaling increased prediction instability. To address these challenges, we introduce Uncertainty-Based Extensible-codebook Federated Learning (UEFL), a novel methodology that explicitly distinguishes between data distributions to improve both accuracy and uncertainty.

Specifically, our design features an advanced codebook comprising a predetermined number of latent vectors (*i.e.* codewords), and employs a discretizer to assign encoded image features to their closest codewords. These codewords, acting as latent representations, are passed to subsequent layers for processing. The codewords are dynamically trained to align with the latent features generated by the image encoder. To mitigate performance degradation when integrating local models from data silos with varying distributions, we initialize a small, shared codebook for all clients. Additional specific codewords are then introduced for individual client use, ensuring explicit differentiation between them. Since the initial codebook is small and requires only a few extensions, the final size remains compact, minimizing the associated computational overhead. Given the privacy constraints in FL, which restrict direct data access, we incorporate an uncertainty evaluator using Monte Carlo Dropout. This evaluator identifies data from diverse distributions, marked by high uncertainty. During training, UEFL systematically distinguishes between these varied distributions and dynamically adds new codewords to the codebook until all distributions are sufficiently represented. In the initial training cycle, shared codewords are randomly initialized. However, in subsequent cycles, the fully trained image encoder is leveraged to initialize new codewords using K-means, aligning them more closely with the data distribution and facilitating faster adaptation to various distributions. As a result, our UEFL model can accommodate data from previously unseen distributions with fewer communication rounds, making it applicable for enhancing other FL algorithms.

To summarize, our contributions are as follows:

- We identify a significant increase in model uncertainty across silos with diverse data distributions in FL, highlighting the challenge of data heterogeneity.

- To address this heterogeneity, we introduce an extensible codebook approach that distinguishes between data distributions by stepwise mapping them to distinct, trainable latent vectors (*i.e.* codewords). This methodology allows for efficient initialization of newly added codewords using a K-means algorithm, closely aligning with the training data feature distributions and

enabling rapid convergence during codebook training.

- We propose a novel data-driven FL framework, named Uncertainty-Based Extensible-codebook Federated Learning (UEFL), which merges the extensible codebook with an uncertainty evaluator. This framework iteratively identifies data from diverse distributions by assessing uncertainty without requiring direct data access. It then processes this data by initializing new codewords to complement the existing codebook, ensuring that each iteration focuses on training the expandable codebook, thus allowing UEFL to adapt seamlessly to new data distributions.

- Our empirical evaluation across various datasets demonstrates that our approach significantly reduces uncertainty by 38.83%-96.24% and enhances model accuracy by 3%-22.1%, evidencing the effectiveness of UEFL in managing data heterogeneity in FL.

## 2. Related Work

### 2.1. Federated Learning

Federated learning (Konečný et al., 2016; Geyer et al., 2017; Chen et al., 2018; Hard et al., 2018; Yang et al., 2019; Ghosh et al., 2020) represents a cutting-edge distributed learning paradigm, specifically designed to exploit data and computational resources across edge devices. The Federated Averaging (FedAvg) algorithm (McMahan et al., 2017), introduced to address the challenges of unbalanced and non-IID data, optimizes the trade-off between computation and communication costs by reducing the necessary communication rounds for training deep networks. FL faces numerous statistical challenges, with data heterogeneity being one of the most critical. In real-world applications, data collected across different clients often varies significantly in terms of distribution, feature space, and sample sizes.

Several methodologies (Zhao et al., 2018; Li et al., 2018; 2019; Kalra et al., 2023) have been developed to address this pivotal issue. PMFL (Zhang et al., 2022) approaches the heterogeneity challenge by drawing inspiration from meta-learning and continual learning, opting to integrate losses from local models over the aggregation of gradients or parameters. DisTrans (Yuan et al., 2022) enhances FL performance through train and test-time distributional transformations, coupled with a novel double-input-channel model architecture. Meanwhile, FCCL (Huang et al., 2022) employs knowledge distillation during local updates to facilitate the sharing of inter and intra domain insights without compromising privacy, and utilizes unlabeled public data to foster a generalizable representation amidst domain shifts. Additionally, the discrete approach to addressing heterogeneity by (Liu et al., 2021), provide further inspiration and valuable perspectives for our research endeavors.

## 2.2. Uncertainty

Recently, the study of uncertainty modeling has gained significant prominence across various research fields, notably within the machine learning community (Chen et al., 2014; Blundell et al., 2015; Kendall & Gal, 2017; Louizos & Welling, 2017; Lahlou et al., 2021; Nado et al., 2021; Gawlikowski et al., 2021). This surge in interest is driven by the critical need to understand and quantify the inherent ambiguity in complex datasets. Techniques such as Monte Carlo Dropout (Gal & Ghahramani, 2016), which introduces variability in model outputs through the use of dropout layers, and Deep Ensembles (Lakshminarayanan et al., 2017), which leverages multiple models with randomly initialized weights trained on identical datasets to evaluate uncertainty, exemplify the advancements in this area. Furthermore, the application of uncertainty modeling has extended beyond traditional domains, impacting fields such as healthcare (Dusenberry et al., 2020) and continual learning (Ahn et al., 2019).

## 3. Methodology

### 3.1. Overall Architecture

Figure 2 illustrates the workflow of our UEFL. Consider multiple data distributions $\mathcal{D}_1, \mathcal{D}_2, ..., \mathcal{D}_M$, with data samples $x \in \mathbb{R}^{H \times W \times D}$, where $H$, $W$, and $D$ denote the input image's height, width, and channel count, respectively, drawn from these $M$ distributions. Upon distributing the global model to local clients, data samples undergo local encoding via a shared encoder $\theta_E$ into feature representations $z \in \mathbb{R}^{h \times w \times d}$, with $h$, $w$, and $d$ representing the features' shape. Subsequently, these features are reshaped into vectors $z \in \mathbb{R}^{l \times d}$, where $l$ is the number of tokens, and divided into $s$ segments $z_i \in \mathbb{R}^{l \times \frac{d}{s}}, \forall i$, with $s$ indicating the segment count. Each segment is mapped to the closest codeword in the codebook via a discretizer $\theta_D$, then reassembled into complete vectors for classification. The classifier $\theta_C$ then deduces the class for the input data, completing the forward processing sequence as follows,

$$z = f_{\theta_E}(x), \quad c = f_{\theta_D}(z), \quad p = f_{\theta_C}(c) \quad (1)$$

where $x$, $z$, $c$, and $p$ denote input data, latent features, discrete coded vectors, and the model prediction, separately.

After loss calculation, models undergo local updates through backpropagation. In a manner akin to FedAvg (McMahan et al., 2017), these updated models are then relayed back to the server for a global update.

$$\theta_k \leftarrow \theta - \eta g_k, \ \forall k \quad (2)$$

$$\theta \leftarrow \sum_{k=1}^{K} \frac{n_k}{n} \theta_k, \quad (3)$$

where $\theta$ denots the global model parameters, $\theta_k$ is the $k$th local model parameters, $g_k$ is the $k$th model gradients, $n_k$ is the number of samples for data silo $k$, and $n$ is the total number of samples for all $K$ silos.

At the end of each iteration, assessing uncertainty through Monte Carlo Dropout is essential, given the privacy constraints of FL, which limit direct access to client data. By evaluating uncertainty against a pre-established threshold, we identify data from heterogeneous distributions. When such data are detected, we augment the codebook with $v$ new codewords initialized by K-means. These newly generated codewords are then exclusively accessible to the corresponding heterogeneous clients, updating the codebook size for the $k$th client from $v_k$ to $v_k + v$, as described in Algorithm 1. This process leverages the fully adapted encoder from previous iterations, utilizing K-means to ensure the new codewords are closely aligned with the actual data distribution, thereby facilitating faster convergence during training. Additionally, since the extended codewords are specific to individual client data and are not included in the integration with other local models, our method ensures that latent features from different distributions remain explicitly differentiated. Consequently, the global model performs better after integration, effectively handling data heterogeneity.

### 3.2. Extensible Codebook

While discretization mitigates data heterogeneity (see Appendix A for theoretical analysis), it struggles with distributions exhibiting significant variations. To effectively handle such heterogeneous data, we design an extensible codebook, beginning with a minimal set of codewords and progressively enlarging this set through a superior initialization strategy that benefits from our UEFL framework. This strategy facilitates stepwise mapping of diverse data distributions to distinct codewords. Starting with a larger codebook can introduce uncertainty in codeword selection due to the concurrent training of multiple codewords.

Similar to VQ-VAE (Van Den Oord et al., 2017), we employ latent vectors as codewords, initializing a compact shared codebook with $v$ codewords $c \in \mathbb{R}^{v \times \frac{d}{s}}$, where $v$ represents the size of the initial codebook. The codewords are initialized using a Gaussian distribution and shared across all data silos. After each iteration's uncertainty assessment, we determine which silos require additional codewords to improve prediction accuracy, and we extend the codebook accordingly for these silos by adding $v$ more codewords. The newly added codewords are initialized using K-means, leveraging the encoder's improved latent features from the prior iteration to better align with the underlying data distribution. To optimize codebook usage, data silos that demonstrated lower performance in the previous iteration are allowed to select codewords from both the newly added codewords and

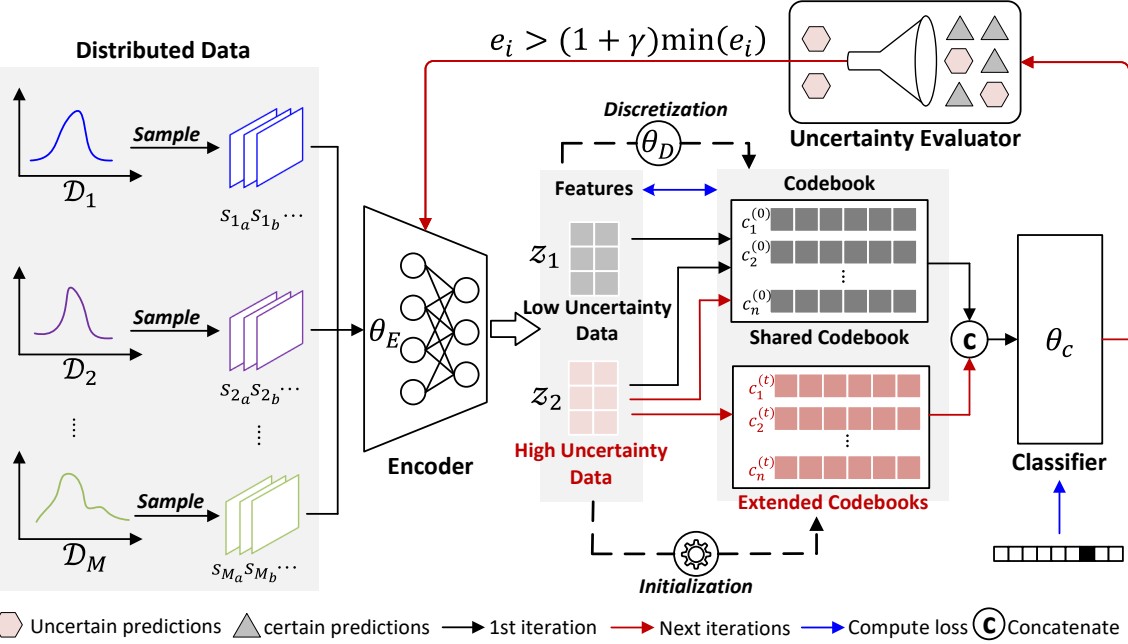

Figure 2: **UEFL flowchart.** In the first iteration, all latent are mapped to initialized shared codewords by the discretizer $\theta_D$. In the next iterations, UEFL identifies data from heterogeneous distributions with the uncertainty evaluator, and complements new codewords with K-means initialization to enhance the codebook. Clients with high uncertainty can select not only newly added codewords but also shared codewords.

the original shared codebook. Typically, the codebook only requires 1-3 extensions until all clients reach low uncertainty levels. The server updates the codebook by computing the average of codewords across the clients that utilize those specific codewords.

For a given iteration, if the codebook size for the $k$th client is $v_k$, the feature vector $z$ is associated with a codeword $c_i$ by the discretizer, which computes the distance between $z$ and all available codewords, selecting the nearest one as follows,

$$i = \underset{j \in 1,2,...,v_k}{\arg\min} ||z - c_j||_2 \qquad (4)$$

**K-means Initialization.** After the first iteration, the adapted encoder produces image features that more accurately reflect the distribution of the training data. Instead of relying on random initialization methods like Gaussian distribution, we initialize new codewords using the centroids of these features, obtained through K-means clustering. This approach expedites codebook training by providing a more informed starting point for the new codewords, allowing them to better align with the underlying data structure. As a result, this initialization strategy facilitates faster convergence and improves the model's ability to adapt to varying data distributions across silos. This strategy hugely reduces the number of training rounds required for model convergence (Details in Appendix J).

**Segmented Codebooks.** For complex datasets, a finite set of discrete codewords might not fully capture the diversity of image features. To bolster the robustness of our methodology, we dissect features into smaller segments to pair them with multiple codewords, thus covering the entirety of a feature vector. This segmentation exponentially increases the codeword pool, ensuring a robust representation capacity without necessitating a large-scale increase and permitting efficient K-means-based initialization. This design minimizes runtime overhead associated with larger codebooks.

### 3.3. Loss Function

Since we introduce learnable codewords in our method, there are two parts of the loss function. For our task, we utilize cross-entropy as the loss function. For codebook optimization, akin to the strategy employed in VQ-VAE (Van Den Oord et al., 2017), we apply a stop gradient operation for the codeword update as follows:

$$\mathcal{L}_{code} = ||sg(c) - z||_2^2 + \beta||c - sg(z)||_2^2 \qquad (5)$$

where $z$ is the image latent features, $c$ is discrete codewords, $\beta$ is a hyper-parameter to adjust the weights of two losses and $sg(\cdot)$ denotes the stop gradient function.

The total loss $\mathcal{L}_{UEFL}$ is the summation of $\mathcal{L}_{task}$ and $\mathcal{L}_{code}$.

**Algorithm 1** Uncertainty-Based Extensible-Codebook Federated Learning (UEFL)

---

**Input:** data distributions $\mathcal{D}_1, \mathcal{D}_2, ..., \mathcal{D}_M$
**Parameters:** uncertainty threshold $\gamma$, learning rate $\eta$, codewords loss weight $\beta$
Sample K data silos from $\mathcal{D}_1, \mathcal{D}_2, ..., \mathcal{D}_M$ as clients
Randomly initialize model parameters $\theta$ and codebook with $v$ codewords
Initially assign uncertainty for all clients to be zero: $e_k = 0, \forall k$
**repeat**
   **for** each round t = 1, 2, ... **do**
      Broadcast $\theta$ to all clients
      **for** all K clients **in parallel do**
         **if** $e_k > (1 + \gamma) \min_{\forall j \in 1,2,...,K}(e_j)$ **then**
            K-means initialize another $v$ codewords and add them to codebook
            Update accesible codewords size for clients with high uncertainty: $v_k \leftarrow v_k + v$
         **end if**
         Encode input into latent features: $z = f_{\theta_E}(x)$
         Discretize latent features to codewords $c_i$, where $i = argmin_{j \in 1,2,...,v_k}||z - c_j||_2$
         Predict with coded vectors: $p = f_{\theta_C}(c_i)$
         Compute codewords loss: $\mathcal{L}_{code} = ||sg(c_i) - z||_2^2 + \beta||c_i - sg(z)||_2^2$
         Compute output loss: $\mathcal{L}_{task} = -\sum y \log p$
         Update local parameters with gradient descent: $\theta_k \leftarrow \theta_k - \eta \nabla_\theta(\mathcal{L}_{code} + \mathcal{L}_{task})$
      **end for**
      Clients return all local models $\theta_k$ to the server
      Update the server model $\theta \leftarrow \sum_{k=1}^{K} \frac{n_k}{n} \theta_k$
   **end for**
   Evaluate uncertainty for each client with integrated model: $e_k = \sum p \log p$
   Reduce the number of communication rounds
**until** $e_k \leq (1 + \gamma) \min_{\forall j \in 1,2,...,K}(e_j), \forall k$

---

### 3.4. Uncertainty Evaluation

As outlined in Section 3.1, evaluating model uncertainty is crucial for identifying data from heterogeneous distributions requiring supplementary codewords. In our work, we utilize Monte Carlo Dropout (MC Dropout) (Gal & Ghahramani, 2016) for uncertainty evaluation, incorporating two dropout layers into our model for regularization purposes. Unlike traditional usage where dropout layers are disabled during inference to stabilize predictions, we activate these layers during testing to generate a variety of outcomes for uncertainty analysis. This variability is quantified using predictive entropy, as described in Equation (6), which serves to measure the prediction dispersion across different evaluations

effectively.

$$e = -\sum_{class} p \log p \qquad (6)$$

A low predictive entropy value signifies model confidence, whereas a high value indicates increased uncertainty. For high entropy, introducing new codewords and conducting additional training rounds are essential steps. Given the variability of uncertainty across datasets, establishing a fixed threshold is impractical. Instead, by analyzing all uncertainty values, we can benchmark against either the minimum or mean values to pinpoint target silos. Our experiments showed superior results when using the minimum value as a reference, thus guiding us to adopt the following threshold criterion:

$$e_k \leq (1 + \gamma) \min_{\forall j \in 1,2,...,K}(e_j), \forall k \qquad (7)$$

where $\gamma$ is a hyperparameter to be set.

Uncertainty decreases consistently during training, making it an effective stopping criterion for codebook extension. Besides, we manually set the maximum number of iterations to 5. A comparison with Deep Ensembles (Lakshminarayanan et al., 2017) is provided in Appendix E.

## 4. Experimental Results

**Experimental Setup.** As discussed in (Kairouz et al., 2021b; Zhou et al., 2023), there are two predominant forms of data heterogeneity in federated learning: feature heterogeneity and label heterogeneity. Our UEFL focuses on tacking feature heterogeneity, and we mainly discuss feature heterogeneity in this section. The discussion for label heterogeneity and the comparison with VHL (Tang et al., 2022), FedBR (Guo et al., 2023b) are in the Appendix I.

Similar to Rotated MNIST (Ghifary et al., 2015), which creates six domains through counter-clockwise rotations of 0°, 15°, 30°, 45°, 60°, and 75° on MNIST, we employ similar technique to introduce feature heterogeneity on five different datasets: MNIST, FMNIST, CIFAR10, GTSRB, and CIFAR100, to validate our framework's robustness. In our experiments, we create three domains by counter-clockwise rotating the datasets by 0° ($\mathcal{D}_1$), -50° ($\mathcal{D}_2$), and 120° ($\mathcal{D}_3$). We sampled three data silos from each domain (*i.e.* totally 9 silos), and data silos for CIFAR100 contain 4000 images each, while the other datasets consist of 2000 images per silo. Besides the regular training with multi-domain data silos, we also test UEFL for domain generalization (DG) task on Rotated MNIST (Ghifary et al., 2015) and PACS (Li et al., 2017) datasets, which contains four distinct domains: art painting (A), cartoon (C), photo (P), and sketch (S).

For RGB datasets like GTSRB, CIFAR10, and CIFAR100, we adopt a pretrained VGG16 model in multi-domain training. In contrast, for grayscale datasets such as MNIST and

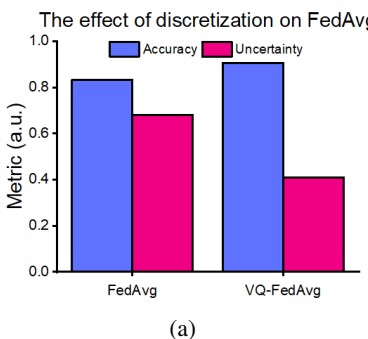
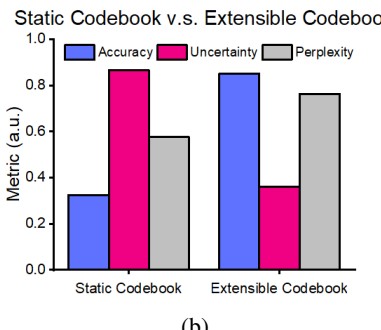
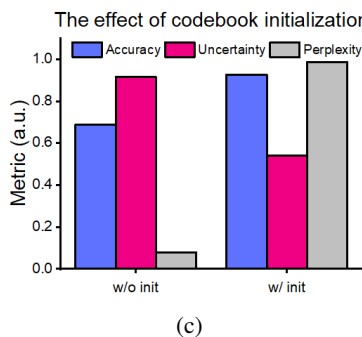

(a)            (b)            (c)

Figure 3: **Design of extensible codebook.** (a) With the discretization (VQ-FedAvg), both accuracy and uncertainty get improved. (b) Our extensible codebook which starts from a small capacity performs better than the static large codebook. (c) With K-means initialization, the utility of codewords (*i.e.* perplexity) gets significantly improved.

Table 1: **UEFL outperforms all baselines on heterogeneous data.** DisTrans lacks a Dropout layer, rendering it incapable of evaluating uncertainty. More results are in Appendix C.

| Methods | MNIST | | FMNIST | | GTSRB | | CIFAR10 | | CIFAR100 | |
|---|---|---|---|---|---|---|---|---|---|---|
| | mA | mE | mA | mE | mA | mE | mA | mE | mA | mE |
| FedAvg (McMahan et al., 2017) | 0.780 | 0.273 | 0.803 | 0.273 | 0.660 | 0.636 | 0.617 | 0.177 | 0.088 | 1.91 |
| DisTrans (Yuan et al., 2022) | 0.815 | - | 0.707 | - | 0.898 | - | 0.699 | - | 0.267 | - |
| **UEFL** (Ours) | **0.920** | **0.149** | **0.850** | **0.167** | **0.942** | **0.0239** | **0.720** | **0.0222** | **0.326** | **0.655** |

FMNIST, lacking pretrained models, we design a convolutional network comprising three ResNet blocks, training it from scratch. And for DG, we adopt a pretrained ResNet18 for both datasets. Initial codebook sizes are set to 32 for MNIST and 64 for the remaining datasets, with an equivalent number of codewords added in each subsequent iteration. While additional iterations may converge within 5 rounds, we extend this to 20 for enhanced experimental clarity. The uncertainty evaluation is conducted 20 times using a dropout rate of 0.1, with thresholds $\gamma$ set at 0.3 for MNIST, 0.1 for FMNIST, GTSRB, and CIFAR100, and 0.2 for CIFAR10, to fine-tune performance. These experiments are performed on a machine with two NVIDIA A6000 GPUs.

**Evaluation Metrics.** We compute the mean Top-1 accuracy (mA), mean entropy (mE), and mean perplexity (mP) across all clients to facilitate a direct comparison, where perplexity quantifies the utility of codewords.

### 4.1. Extensible Codebook

**Discretization for Heterogeneous FL.** To show the effectness of discretization to tackle the data heterogeneity in FL, we design a toy experiment on MNIST. Temporarily setting aside federated learning's privacy considerations, we directly discretized the features for each client using the distinct codebooks based on its originating domain. With this discretization of VQ-FedAvg, the mean accuracy was improved from 0.834 to 0.907 with the reduction of uncertainty,

demonstrating the effectiveness of feature discretization in enhancing performance within a heterogeneous federated learning context, as shown in Figure 3(a).

**Extensible Codebook v.s. Static Large Codebook.** To validate our extensible codebook's superiority over starting with a large codebook, we ensured both methods ended with the same number of codewords through experiments. Results on CIFAR100 in Figure 3(b) demonstrate the difficulties associated with a larger initial codebook in codeword selection for image features. Conversely, gradually expanding the codebook significantly improved codeword differentiation, yielding better outcomes, such as enhanced accuracy (from 0.13 to 0.34), reduced uncertainty (0.78 vs. 1.66 for the static approach), and increased utilization of codewords.

**Codebook Initialization.** Section 3.1 highlights our UEFL's capability for efficient codeword initialization via K-means, utilizing features from a finetuned encoder. The efficacy of initialization is validated in Figure 3(c) with results on MNIST, showing enhancements across all metrics.

### 4.2. UEFL for Multi-Domain Learning

We conducted comparative experiments on five datasets with introduced feature heterogeneity against leading algorithms, specifically the baseline FedAvg (McMahan et al., 2017) and DisTrans (Yuan et al., 2022). For accuracy comparison, DisTrans generally exhibits better performance than FedAvg, making it our primary point of comparison. Re-

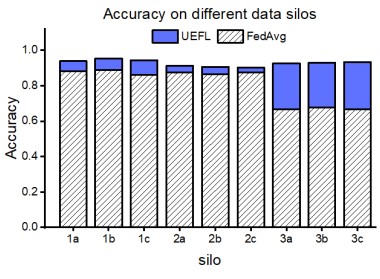
(a) Accuracy comparison.

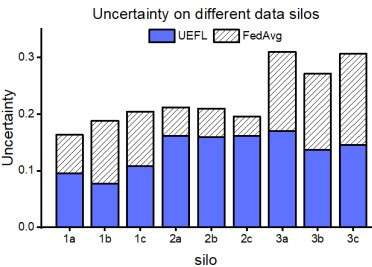
(b) Uncertainty comparison.

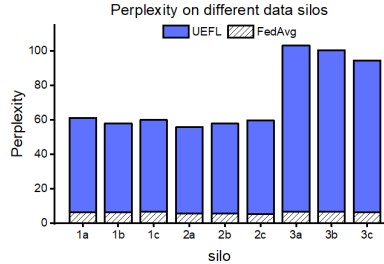
(c) Perplexity comparison.

Figure 4: Experiments are on MNIST. $\mathcal{D}_3$ presents much lower accuracy and higher uncertainty compared to $\mathcal{D}_1$, $\mathcal{D}_2$ for FedAvg. And the perplexity results show that our UEFL assigns new codewords to $\mathcal{D}_3$ to improve the performance.

Table 2: **Comparison with different methods for DG.** Results are on six domains of Rotated MNIST, four domains of PACS and their average. Our approach is compared with baselines: FedAvg, FedSR, and FedIIR.

| Methods | Rotated MNIST | | | | | | | PACS | | | | |
|---|---|---|---|---|---|---|---|---|---|---|---|---|
| | $\mathcal{M}_0$ | $\mathcal{M}_{15}$ | $\mathcal{M}_{30}$ | $\mathcal{M}_{45}$ | $\mathcal{M}_{60}$ | $\mathcal{M}_{75}$ | **Ave.** | A | C | P | S | **Ave.** |
| FedAvg (McMahan et al., 2017) | 82.7 | 98.2 | 99 | 99.1 | 98.2 | 89.9 | 94.5 | 78 | 73 | 92 | 79 | 80.3 |
| FedSR (Nguyen et al., 2022) | 84.2 | 98.0 | 98.9 | 99.0 | 98.3 | 90.0 | 94.7 | 83 | 75 | 94 | 82 | 83.4 |
| FedIIR (Guo et al., 2023a) | 83.8 | 98.2 | 99.1 | 99.1 | 98.5 | 90.8 | 95.0 | 83 | 76 | 94 | 82 | 83.7 |
| **UEFL** (ours) | 88.1 | 97.3 | 97.6 | 97.8 | 97.9 | 93.2 | **95.3** | 81 | 80 | 94 | 82 | **84.5** |

garding uncertainty comparison, because DisTrans lacks Dropout layers, precluding uncertainty evaluation, we exclusively compare uncertainty metrics with FedAvg.

**Performance.** The results in Table 1 provide a comprehensive comparison, illustrating that UEFL surpasses all other SOTA methods in both accuracy and uncertainty reduction. Specifically, UEFL achieves accuracy improvements ranging from 3% to 22.1% over DisTrans. Our UEFL improves uncertainty compared to FedAvg, achieving reductions by 38.83%-96.24%. Figures 4(a) and 4(b) details performance across each data silo, highlighting UEFL's effectiveness in elevating the accuracy and degrading the uncertainty.

**Codewords Perplexity.** Figure 4(c) presents a perplexity comparison between our UEFL and FedAvg, illustrating enhanced codebook utilization after assigning new codewords to $\mathcal{D}_3$. This adjustment not only benefits $\mathcal{D}_3$ but also improves the codebook utilization for $\mathcal{D}_1$ and $\mathcal{D}_2$.

**Computation Overhead.** Our approach introduces only a small codebook, thus incurring negligible memory and computational overheads. Specifically, for the CIFAR10 dataset, the parameter count for the baseline FedAvg model is 14.991M, whereas our UEFL model slightly increases to 15.491M, indicating a tiny memory increment of 3.34%. In terms of runtime, UEFL also exhibits a minimal increase from 16.154ms to 16.733ms (3.58% increase). These findings underscore UEFL's suitability for deployment on edge devices. More details are included in Appendix G.

### 4.3. UEFL for Domain Generalization (DG)

For DG task, the trained model needs to be evaluated on an out-of-distribution domain and we follow the evaluation method in (Nguyen et al., 2022; Guo et al., 2023a). Specifically, we perform "leave-one-domain-out" experiments, where we choose one domain as the target domain, train the model on all remaining domains, and evaluate it on the chosen domain. Each source domain is treated as a client.

As shown in Table 2, our UEFL enhanced mean accuracy on the RotatedMNIST dataset, elevating it from 0.945 to 0.953. This performance exceeds that of FedSR (Nguyen et al., 2022) at 0.947 and FedIIR (Guo et al., 2023a) at 0.95. Similarly, on the PACS dataset, UEFL improved mean accuracy from 0.803 to 0.8453, surpassing FedSR's 0.834 and FedIIR's 0.837. These results underscore UEFL's efficacy in tackling feature heterogeneity and superior performance on the federated DG task, beating state-of-the-art methods.

### 4.4. Ablation Study

**Imbalanced Clients.** We constructed an experimental setup with three data silos from $\mathcal{D}_1$ and one each from $\mathcal{D}_2$ and $\mathcal{D}_3$, totaling five silos. Our UEFL can also improve both accuracy (from 0.508 to 0.828) and uncertainty (from 0.256 to 0.105) in this scenario. Detailed results are in Appendix K.

**Large Number of Clients.** We follow (Guo et al., 2023a) to further partition the five training domains of Rotated MNIST into 50 sub-domains, and adopt the setup from (Zhang et al.,

Table 3: UEFL is scalable for 50 clients and beats all SOTA methods: FedAvg, FedSR and FedIIR.

| Methods | #Clients | Backbone | Domains | | | | | | Average |
|---|---|---|---|---|---|---|---|---|---|
| | | | $\mathcal{M}_0$ | $\mathcal{M}_{15}$ | $\mathcal{M}_{30}$ | $\mathcal{M}_{45}$ | $\mathcal{M}_{60}$ | $\mathcal{M}_{75}$ | |
| FedAvg (McMahan et al., 2017) | 50 | ResNet18 | 77.9 | 95.9 | 96.9 | 97 | 96 | 81.2 | 90.8 |
| FedSR (Nguyen et al., 2022) | 50 | ResNet18 | 78.3 | 95.7 | 96.3 | 97.1 | 96 | 84 | 91.2 |
| FedIIR (Guo et al., 2023a) | 50 | ResNet18 | 84 | 96.8 | 97.7 | 97.7 | 97.4 | 84.5 | 93 |
| **UEFL** (ours) | 50 | ResNet18 | 86.4 | 95.5 | 96.4 | 96.9 | 94.7 | 90.6 | **93.42** |

Table 4: UEFL is also scalable for 100 clients and beats all SOTA methods following the experimental setup in FedCR.

| Methods | #Clients | EMNIST-L | FMNIST | CIFAR10 | CIFAR100 |
|---|---|---|---|---|---|
| FedAvg (McMahan et al., 2017) | 100 | 95.89 | 88.15 | 76.83 | 32.08 |
| FedSR (Nguyen et al., 2022) | 100 | 86.22 | 85.55 | 61.47 | 40.82 |
| FedCR (Zhang et al., 2023) | 100 | 97.47 | 93.78 | 84.74 | 62.96 |
| **UEFL** (ours) | 100 | **98.29** | **93.93** | **86.11** | **63.37** |

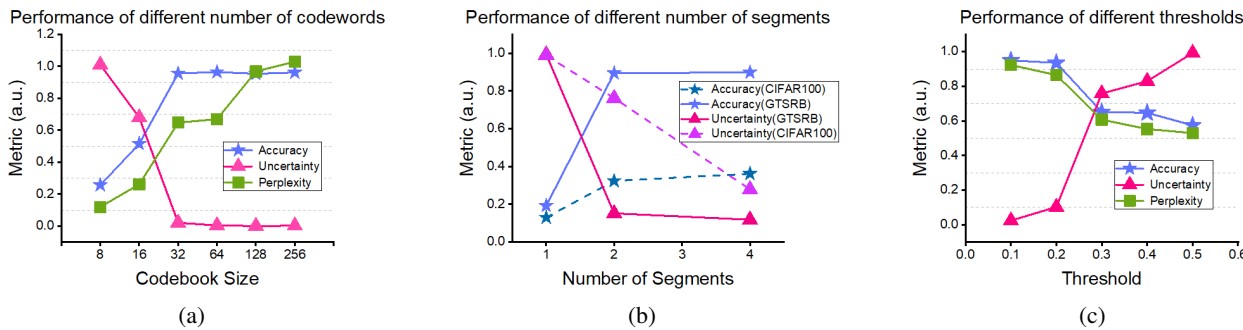

Figure 5: (a) 32 codewords are sufficient. (b) CIFAR100 requires 4 segments. (c) A smaller threshold performs better.

2023) to evaluate performance on 100 clients. In both cases, UEFL achieves the highest mean accuracy, outperforming all baselines, as shown in Table 3 and Table 4, demonstrating its scalability to a larger number of clients.

**Number of codewords and segments.** We investigate the impact of varying the number of initialized codewords in our extensible codebook, to balance accuracy with runtime efficiency in K-means initialization. In Figure 5(a) for GTSRB, initializing with 32 codewords provides comparable accuracy and uncertainty metrics. For more complex datasets, we enhance selection capacity using codeword segmentation. Figure 5(b) demonstrates that segmenting codewords into 4 parts leads to enhanced performance on CIFAR100.

**Uncertainty Threshold.** In our UEFL, the uncertainty evaluator plays a pivotal role in identifying heterogeneous data without needing direct data access, with the threshold selection being critical. As illustrated in Figure 5(c), a lower threshold imposes stricter criteria, pushing the model to achieve higher performance. However, it's important to recognize that beyond a certain point, further reducing the threshold may not significantly enhance outcomes but will

increase computational overhead. Thus, in such cases, there is a trade-off between runtime and performance.

## 5. Conclusion

In this work, we address the challenge of data heterogeneity among silos within federated learning setting by introducing an innovative solution: an extensible codebook designed to map distinct data distributions using varied codeword pools. Our proposed framework, Uncertainty-Based Extensible-Codebook Federated Learning (UEFL), leverages this extensible codebook through an iterative process that adeptly identifies data from unknown distributions via uncertainty evaluation and enriches the codebook with newly initialized codewords tailored to these distributions. The iterative nature of UEFL, coupled with efficient codeword initialization using K-means, ensures codewords are closely matched with the actual data distribution, thereby expediting model convergence. This approach allows UEFL to rapidly adjust to new and unseen data distributions, enhancing adaptability. Our comprehensive evaluation across various prominent datasets showcases UEFL's effectiveness.

## Acknowledgements

This work was supported in part by the joint funding program between the National University of Singapore and the University of Toronto. We gratefully acknowledge their support.

## Impact Statement

This paper presents work whose goal is to advance the field of Machine Learning. There are many potential societal consequences of our work, none which we feel must be specifically highlighted here.

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

## A. Discretization for Generalization

Theoretically, the inclusion of the discretization process offers two key advantages: (1) enhanced noise robustness, and (2) reduced underlying dimensionality. These benefits are demonstrated in the following two theorems. (Liu et al., 2021).

**Notation:** $h$ is input vector, $h \in \mathcal{H} \in \mathcal{R}^m$. $L$ is the size of codebook, $G$ is the number of segments, $q(\cdot)$ is discretization process, $\phi(\cdot)$ is any function (model). Given any family of sets $S = \{S_1, ..., S_K\}$ with $S_1, ..., S_K \subseteq \mathcal{H}$, we define $\phi_k^S$ by $\phi_k^S = \mathbb{1}\{h \in S_k\}\phi(h)$ for all $k \in [K]$, where $[K] = \{1, ..., K\}$. And we denote by $(Q_k)_{k \in [L^G]}$ all the codewords.

**Theorem 1:** (with discretization) *Let $S_k = \{Q_k\}$ for all $k \in [L^G]$. Then, for any $\delta > 0$, with probability at least $1 - \delta$ over an iid draw of $n$ examples $(h_i)_{i=1}^n$, the following holds for any $\phi : \mathcal{R}^m \to \mathcal{R}$ and all $k \in [L^G] :$ if $|\phi_k^S(h)| \leq \alpha$ for all $h \in \mathcal{H}$, then*

$$\left| \mathbb{E}_h[\phi_k^S(q(h, L, G))] - \frac{1}{n}\sum_{i=1}^n \phi_k^S(q(h_i, L, G)) \right| = \mathcal{O}\left(\alpha\sqrt{\frac{G\ln(L) + \ln(2/\delta)}{2n}}\right), \qquad (8)$$

*where no constant is hidden in $\mathcal{O}$.*

**Theorem 2:** (without discretization) *Assume that $||h||_2 \leq R_{\mathcal{H}}$ for all $h \in \mathcal{H} \in \mathcal{R}^m$. Fix $\mathcal{C} \in argmin_{\overline{\mathcal{C}}}\{|\overline{\mathcal{C}}| : \overline{\mathcal{C}} \subseteq \mathcal{R}^m, \mathcal{H} \subseteq \cup_{c \in \overline{\mathcal{C}}}\mathcal{B}[c]\}$ where $\mathcal{B}[c] = \{x \in \mathcal{R}^m : ||x - c||_2 \leq R_{\mathcal{H}}/(2\sqrt{n})\}$. Let $S_k = \mathcal{B}[c_k]$ for all $k \in [|\mathcal{C}|]$ where $c_k \in \mathcal{C}$ and $\cup_k\{c_k\} = \mathcal{C}$. Then, for any $\delta > 0$, with probability at least $1 - \delta$ over an iid draw of $n$ examples $(h_i)_{i=1}^n$, the following holds for any $\phi : \mathcal{R}^m \to \mathcal{R}$ and all $k \in [|\mathcal{C}|] :$ if $|\phi_k^S(h)| \leq \alpha$ for all $h \in \mathcal{H}$ and $|\phi_k^S(h) - \phi_k^S(h')| \leq \varsigma_k||h - h'||_2$ for all $h, h' \in S_k$, for all $h, h' \in S_k$, then*

$$\left| \mathbb{E}_h[\phi_k^S(h)] - \frac{1}{n}\sum_{i=1}^n \phi_k^S(h_i) \right| = \mathcal{O}\left(\alpha\sqrt{\frac{m\ln(4\sqrt{nm}) + \ln(2/\delta)}{2n}} + \frac{\overline{\varsigma_k}R_{\mathcal{H}}}{\sqrt{n}}\right), \qquad (9)$$

*where no constant is hidden in $\mathcal{O}$ and $\overline{\varsigma_k} = \varsigma_k(\frac{1}{n}\sum_{i=1}^n \mathbb{1}\{h_i \in \mathcal{B}[c_k]\})$.*

Based on these two theorems, we can determine that the performance gap between training and test data is smaller when discretization is applied, due to the following two points:

- There is an additional error without discretization (i.e. $\frac{\overline{\varsigma_k}R_{\mathcal{H}}}{\sqrt{n}}$) in the bound of Theorem 2. This error disappears with discretization in the bound of Theorem 1 as the discretization process reduces the sensitivity to noise.

- The discretization process reduces the underlying dimensionality of $m\ln(4\sqrt{nm})$ without discretization (in Theorem 2) to that of $G\ln(L)$ with discretization (in Theorem 1). Since the number of discretization heads $G$ (eg. $G$ is 1, 2, or 4 in our case) is always much smaller than the number of dimensions $m$, the inequality $G\ln(L) \leq m\ln(4\sqrt{nm})$ consistently holds.

## B. Discretization for non-iid Federated Learning

**Notation:** Consider a federated setting with $K$ clients. Client $k$ has $n_k$ samples independently drawn from its own distribution $P_k$. Total samples of all clients $n = \sum_{k=1}^K n_k$. The global distribution is $\overline{P} = \sum_{k=1}^K \frac{n_k}{n}P_k$.

**Theorem 3:** (with discretization) *if $|\phi_k^S(h)| \leq \alpha$ for all $h \in \mathcal{H}$, then, for any $\delta > 0$, with probability at least $1 - \delta$:*

$$\left| \sum_{k=1}^K \frac{n_k}{n}\mathbb{E}_{h \sim P_k}[\phi_k^S(q(h, L, G))] - \frac{1}{n}\sum_{k=1}^K\sum_{i=1}^{n_k} \phi_k^S(q(h_i^{(k)}, L, G)) \right| = \mathcal{O}\left(\alpha\sqrt{\frac{G\ln L + \ln(2K/\delta)}{2n}} + \frac{\nu^{(q)}}{\sqrt{n}}\right), \qquad (10)$$

*where $\nu^{(q)} = \frac{1}{K}\sum_{k=1}^K \text{Div}(P_k^{(q)}, \overline{P}^{(q)})$, denoting the KL divergence between the client distributions and the global distribution after discretization.*

**Theorem 4:** (without discretization) *Assume that* $||\boldsymbol{h}||_2 \leq R_{\mathcal{H}}$, *then for any* $\delta > 0$, *with probability* $\geq 1 - \delta$,

$$\left| \sum_{k=1}^{K} \frac{n_k}{n} \mathbb{E}_{\boldsymbol{h} \sim P_k}[\phi_k^S(\boldsymbol{h})] - \frac{1}{n} \sum_{k=1}^{K} \sum_{i=1}^{n_k} \phi_k^S(\boldsymbol{h}_i^{(k)}) \right| = \mathcal{O}\left(\alpha \sqrt{\frac{m \ln(4\sqrt{nm}) + \ln(2K/\delta)}{2n}} + \frac{\overline{\varsigma} R_{\mathcal{H}} + \nu}{\sqrt{n}}\right), \quad (11)$$

*where* $\nu = \frac{1}{K} \sum_{k=1}^{K} \mathrm{Div}(P_k, \overline{P})$, *denoting the KL divergence between the client distributions and the global distribution.*

The KL divergence term $\nu^{(q)}$ after discretization is significantly smaller than the original $\nu$ due to discretization-induced invariance. Therefore, discretization not only improves robustness to noise and reduces dimensionality, but importantly, it explicitly and effectively mitigates the adverse effects of data heterogeneity typical in non-IID federated learning environments.

## C. Different data heterogeneity for UEFL

### C.1. UEFL for Multi-Domain Learning

Table 5: **UEFL outperforms all baselines on heterogeneous data.** DisTrans lacks a Dropout layer, rendering it incapable of evaluating uncertainty. CIFAR100$^\star$ exhibits poor performance due to the highly heterogeneous experimental setup.

| Methods | Data | MNIST | | FMNIST | | GTSRB | | CIFAR10 | | CIFAR100$^\star$ | |
|---|---|---|---|---|---|---|---|---|---|---|---|
| | | mA | mE | mA | mE | mA | mE | mA | mE | mA | mE |
| FedAvg | $\mathcal{D}_1$ | 0.874 | 0.212 | 0.801 | 0.246 | 0.670 | 0.623 | 0.676 | 0.172 | 0.110 | 1.74 |
| | $\mathcal{D}_2$ | 0.848 | 0.231 | 0.825 | 0.232 | 0.677 | 0.634 | 0.622 | 0.178 | 0.072 | 1.86 |
| | $\mathcal{D}_3$ | 0.618 | 0.377 | 0.784 | 0.341 | 0.634 | 0.652 | 0.553 | 0.183 | 0.083 | 2.13 |
| | All | 0.780 | 0.273 | 0.803 | 0.273 | 0.660 | 0.636 | 0.617 | 0.177 | 0.088 | 1.91 |
| DisTrans | $\mathcal{D}_1$ | 0.856 | - | 0.721 | - | 0.898 | - | 0.721 | - | 0.289 | - |
| | $\mathcal{D}_2$ | 0.799 | - | 0.705 | - | 0.900 | - | 0.719 | - | 0.261 | - |
| | $\mathcal{D}_3$ | 0.789 | - | 0.694 | - | 0.897 | - | 0.659 | - | 0.251 | - |
| | All | 0.815 | - | 0.707 | - | 0.898 | - | 0.699 | - | 0.267 | - |
| **UEFL** (Ours) | $\mathcal{D}_1$ | 0.951 | 0.120 | 0.857 | 0.147 | 0.95 | 0.0196 | 0.776 | 0.0192 | 0.362 | 0.728 |
| | $\mathcal{D}_2$ | 0.885 | 0.196 | 0.848 | 0.188 | 0.964 | 0.0206 | 0.713 | 0.0245 | 0.335 | 0.624 |
| | $\mathcal{D}_3$ | 0.924 | 0.131 | 0.845 | 0.167 | 0.911 | 0.0314 | 0.671 | 0.0229 | 0.282 | 0.612 |
| | All | **0.920** | **0.149** | **0.850** | **0.167** | **0.942** | **0.0239** | **0.720** | **0.0222** | **0.326** | **0.655** |

### C.2. Lower Data heterogeneity on CIFAR100

In Table 5, CIFAR100 exhibits poor performance due to the highly heterogeneous experimental setup. To provide a detailed comparison with the baseline, we used a VGG16 backbone to test CIFAR100 under multiple settings: (1) Local Training: all data trained together without a distributed setting; (2) FedAvg (w/o hete): CIFAR100 split into 5 clients, each with 10,000 images, to evaluate FedAvg performance; (3) FedAvg (w/ hete): images for the 5 clients were rotated by -30°, -15°, 0°, 15°, and 30°, respectively, to introduce data heterogeneity, and FedAvg performance was evaluated; (4) UEFL (w/ hete): tested under the same heterogeneous setup. We trained models from scratch and with pre-trained weights. Results are presented in Table 6, showing that UEFL consistently outperforms FedAvg in both cases.

### C.3. Different data heterogeneity on CIFAR100

By progressively increasing the rotation angles to simulate greater data heterogeneity, we evaluated UEFL's performance under varying levels of heterogeneity. As shown in Table 7, while overall performance decreases with higher heterogeneity, UEFL consistently outperforms FedAvg, with the performance gap widening as heterogeneity increases, demonstrating UEFL's superiority in addressing data heterogeneity.

Table 6: Under lower data heterogeneity on CIFAR100, UEFL continues to outperform FedAvg for both training from scratch and using pre-trained weights.

| Training Strategy | Local training | FedAvg (w/o hete) | FedAvg (w/ hete) | UEFL (w/ hete) |
|---|---|---|---|---|
| From scratch | 0.3852 | 0.2447 | 0.0852 | 0.1062 |
| Pre-trained | 0.6604 | 0.6496 | 0.5005 | 0.5619 |

Table 7: With different rotation angles, UEFL keep outperforms FedAvg.

| Rotation Angles | FedAvg | UEFL |
|---|---|---|
| {-10°, -5°, 0°, 5°, 10°} | 0.5935 | 0.6232 |
| {-20°, -10°, 0°, 10°, 20°} | 0.5469 | 0.6039 |
| {-30°, -15°, 0°, 15°, 30°} | 0.5106 | 0.56 |
| {-40°, -20°, 0°, 20°, 40°} | 0.4799 | 0.5311 |
| {-50°, -25°, 0°, 25°, 50°} | 0.4494 | 0.5074 |
| {-60°, -30°, 0°, 30°, 60°} | 0.4368 | 0.4939 |
| {-70°, -35°, 0°, 35°, 70°} | 0.4188 | 0.4737 |

## D. Optimal training epochs of baselines

To fully demonstrate the efficacy of our UEFL, besides evaluating baselines with the same total training epochs as UEFL, we remove the additional training epochs from UEFL iterations and obtain the optimal performance, for more fair comparison.

Table 8: UEFL outperforms all baselines without additional training epochs.

| Methods | MNIST | | FMNIST | | GTSRB | | CIFAR10 | | CIFAR100 | |
|---|---|---|---|---|---|---|---|---|---|---|
| | mA | mE | mA | mE | mA | mE | mA | mE | mA | mE |
| FedAvg | 0.782 | 0.261 | 0.801 | 0.289 | 0.657 | 0.645 | 0.618 | 0.173 | 0.093 | 1.74 |
| **UEFL** (Ours) | **0.920** | **0.149** | **0.850** | **0.167** | **0.942** | **0.0239** | **0.720** | **0.0222** | **0.326** | **0.655** |

## E. Deep Ensembles

We evaluated Deep Ensembles (Lakshminarayanan et al., 2017) by creating 5 ensembles to assess uncertainty for our method. Table 9 shows a comparison between Deep Ensembles and Monte Carlo Dropout. From the results, we observe that the accuracy when using Deep Ensembles is quite similar to Monte Carlo Dropout, apart from the stochastic variations. This is expected, as the accuracy is not directly impacted by the choice of uncertainty evaluation method. However, the uncertainty values for Deep Ensembles are higher than those for Monte Carlo Dropout, likely due to the use of only 5 ensembles for evaluation to reduce computational time. In conclusion, while the accuracy is comparable, Deep Ensembles require significantly more computational resources due to the need to train multiple networks. Therefore, Monte Carlo Dropout is a more efficient and suitable choice for our approach.

## F. Neural Collapse

We conducted experiments on MNIST and CIFAR-100 based on the framework presented in (Papyan et al., 2020). According to the paper, when training continues until the training error reaches 0 (*i.e.* training accuracy exceeds 99.9% for MNIST/CIFAR-10), the Terminal Phase of Training (TPT) begins, during which neural collapse (NC) emerges. To validate this, we first conducted local training by training on all data together. Our results confirmed the paper's claim that additional training beyond the zero-error point leads to improved performance. We then extended these experiments to the federated learning setting. The detailed results are presented in Table 10.

Table 9: Comparison of Deep Ensembles and Monte Carlo Dropout for uncertainty evaluation.

| Methods | MNIST | | FMNIST | | GTSRB | | CIFAR10 | | CIFAR100 | |
|---|---|---|---|---|---|---|---|---|---|---|
| | mA | mE | mA | mE | mA | mE | mA | mE | mA | mE |
| Monte Carlo Dropout | 0.920 | 0.149 | 0.850 | 0.167 | 0.942 | 0.0239 | 0.720 | 0.0222 | 0.326 | 0.655 |
| Deep Ensemble | 0.926 | 0.211 | 0.853 | 0.289 | 0.940 | 0.041 | 0.717 | 0.043 | 0.331 | 0.873 |

From these experiments, we observed the following findings: 1. The dropout layer must be removed; otherwise, the training accuracy cannot exceed 99.9% (e.g., the final training accuracy for MNIST is limited to 96% with dropout). 2. Compared to local training, more training epochs are required to reach the TPT in federated learning. For example, on CIFAR-100 with data heterogeneity, federated learning requires 44 rounds of training (44 × 5 epochs), while local training achieves TPT in 38 epochs. 3. Although neural collapse yields improved performance, UEFL consistently outperforms it, especially on more complex datasets like CIFAR-100. This is partly because removing dropout layers for neural collapse increases the risk of overfitting.

Table 10: Comparison with neural collapse (NC).

| Method | MNIST | CIFAR100 |
|---|---|---|
| zero-error | 0.9375 | 0.3473 |
| last epoch (NC) | 0.9584 | 0.3482 |
| UEFL (Ours) | 0.9778 | 0.5074 |

## G. Computation Cost

Table 11 compares FedAvg and UEFL. The parameter count for UEFL in this table represents the final model size, including 256 codewords after two iterations starting from 64. In our experiments, 256 is the largest final codebook size across all datasets. For simpler datasets like MNIST, UEFL demonstrates an even lower computation cost.

Table 11: With different rotation angles, UEFL keep outperforms FedAvg.

| Method | #Params (M) | CPU runtime (ms) | GPU runtime (ms) |
|---|---|---|---|
| FedAvg | 14.991 | 16.102 | 16.154 |
| UEFL (Ours) | 15.491 | 16.611 | 16.733 |

## H. Convergence Curves

For the experiments on the CIFAR10 dataset, in Figure 6, at round 40, after we assign new codewords, rapid performance gains are evident. Remarkably, the training process demonstrates swift convergence, typically within just five rounds. For illustrative clarity and to underscore the differential impact, we extend the training to 20 rounds in subsequent iterations, showcasing the accelerated and effective adaptation of our approach. In addition, after 60 rounds, even if we keep adding new codewords, the increased perplexity denotes a higher utilization of the codebook. However, there is no significant improvement in accuracy or uncertainty. Figure 6 also shows a large boost with our UEFL on MNIST.

For the experiments on the FMNIST and GTSRB datasets, as depicted in Figure 7, we introduce new codewords to the codebook only once. Notably, there is a clear "performance jump" evident in all six figures, showcasing the rapid adaptation of our UEFL to new data distributions.

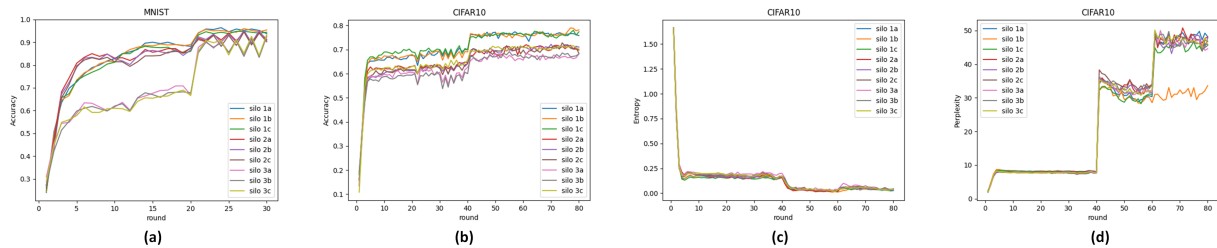

Figure 6: **Learning curves of MNIST and CIFAR10.** (a) Accuracy of MNIST. (b) Accuracy of CIFAR10. (c) Uncertainty of CIFAR10. (d) Perplexity of CIFAR10.

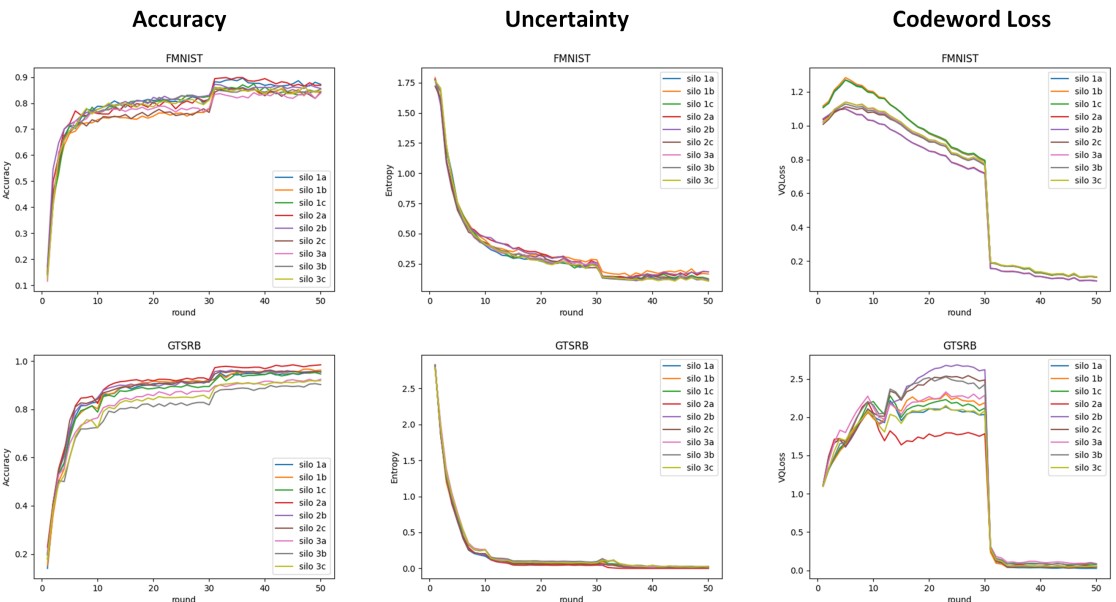

Figure 7: **Learning curves of FMNIST and GTSRB.** (left) Accuracy results. (mid) Uncertainty results. (right) codeword loss results.

## I. Our UEFL for Label Heterogeneity

Similar to (Tang et al., 2022; Guo et al., 2023b), we introduce label heterogeneity with dirichlet distribution ($\alpha = 0.1$). The results in Table 12 show that our UEFL can also tackle the label heterogeneity when compared to FedAvg and performs better than VHL for CIFAR10 even if it cannot perform as well as FedBR.

Table 12: **Comparison with different methods on data with label heterogeneity.**

| Method | FMNIST | | | CIFAR10 | | | |
|---|---|---|---|---|---|---|---|
| | FedAvg | VHL | UEFL (ours) | FedAvg | VHL | FedBR | UEFL (ours) |
| mA | 87.45 | 91.52 | 90.59 | 58.99 | 61.23 | 64.61 | 62.67 |

## J. K-means Initialization

Figure 8 illustrates this concept: gray points represent features from the trained encoder, clustered according to their data distributions. While direct data access is restricted, differentiation by uncertainty allows us to identify and utilize the centroids of these clusters via K-means for codeword initialization.

Table 13: Our UEFL improves the performance of unbalanced data.

| Data | Silo | FedAvg | | UEFL | |
|---|---|---|---|---|---|
| | | Acc | Entropy | Acc | Entropy |
| $\mathcal{D}_1$ | $s_{1_a}$ | 0.964 | 0.0312 | 0.952 | 0.0291 |
| | $s_{1_b}$ | 0.936 | 0.0252 | 0.974 | 0.0170 |
| | $s_{1_c}$ | 0.964 | 0.0499 | 0.944 | 0.0308 |
| $\mathcal{D}_2$ | $s_{2_a}$ | 0.796 | 0.1477 | **0.836** | **0.1261** |
| $\mathcal{D}_3$ | $s_{3_a}$ | 0.508 | 0.2560 | **0.828** | **0.1048** |

And to bolster the robustness of our methodology, we dissect features into smaller segments—using factors like 2 or 4—to pair them with multiple codewords, thus covering the entirety of a feature vector as illustrated in Figure 3. This segmentation exponentially increases the codeword pool to $n^2$ or $n^4$, ensuring a robust representation capacity.

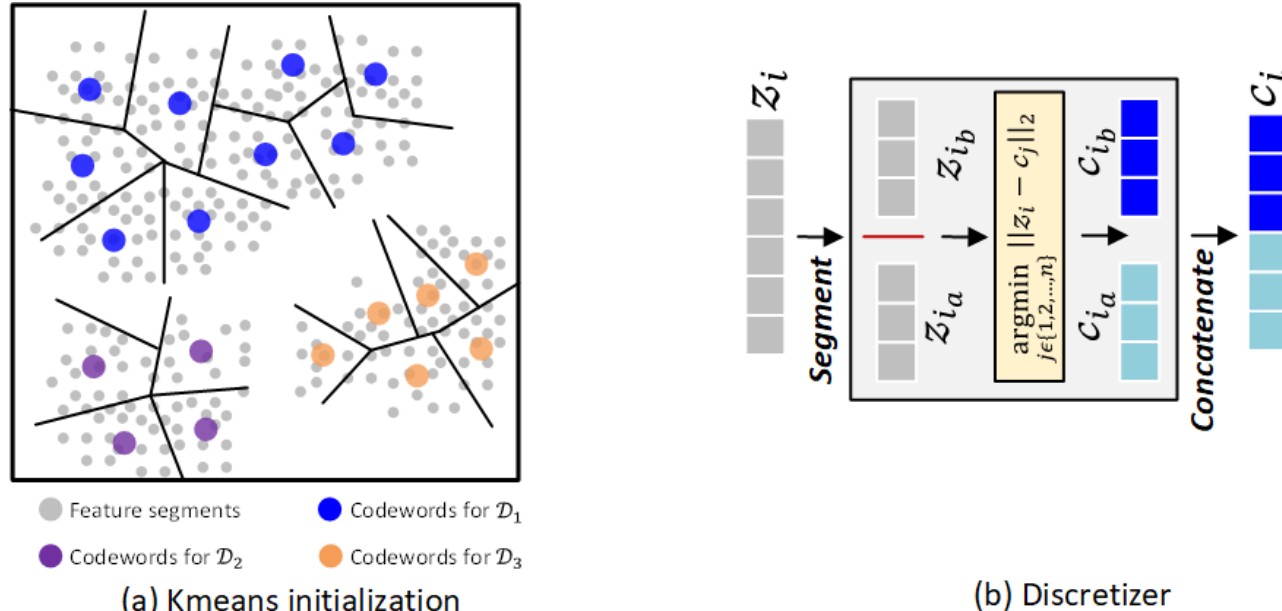

(a) Kmeans initialization    (b) Discretizer

Figure 8: (a) Kmeans initialization for heterogeneous data silos. (b) Workflow of discretizer.

## K. Imbalanced Clients

As shown in Table 13, our UEFL also works for imbalanced data silos when there are three clients sampled from the same domain. Both accuracy and uncertainty get improved, especially for the third domain.

## L. UEFL Optimization

**Number of Codewords.** We investigate the impact of varying the number of initialized codewords within our extensible codebook in Table 14, aiming to strike a balance between achieving competitive accuracy and optimizing the runtime efficiency of the K-means initialization. Our findings, for the GTSRB dataset, reveal that starting with 32 or 64 codewords offers comparable accuracy and uncertainty metrics to larger codebooks, while significantly enhancing the efficiency of the K-means initialization. This efficiency highlights the efficacy of our proposed approach.

In addition, for more complex datasets, requiring a broader representation of image features but with minimal initialization time, we employ codeword segmentation to enhance selection capacity efficiently. We explore the impact of segmentation factors of 1, 2, and 4, starting with 16 codewords for GTSRB and 32 for CIFAR100. Our findings indicate that, particularly

for CIFAR100, splitting vectors into 4 segments with only 32 initialized codewords achieves impressive performance. Similarly, for GTSRB, segmentation into 2 parts is adequate for effective image feature representation.

| #Codes | Data | $\mathcal{L}_{\mathbf{code}}\downarrow$ | mP↑ | mE↓ | mA↑ | codebook growth |
|---|---|---|---|---|---|---|
| 8 | $\mathcal{D}_1$ | 3.61 | 4.78 | 2.02 | 0.257 | |
| | $\mathcal{D}_2$ | 3.68 | 4.86 | 2.01 | 0.265 | $8 \to 16 \to 32 \to 64$ |
| | $\mathcal{D}_3$ | 3.38 | 4.81 | 2.03 | 0.249 | |
| 16 | $\mathcal{D}_1$ | 3.41 | 10.46 | 1.36 | 0.515 | |
| | $\mathcal{D}_2$ | 3.55 | 10.54 | 1.37 | 0.506 | $16 \to 32 \to 64$ |
| | $\mathcal{D}_3$ | 3.23 | 10.61 | 1.39 | 0.486 | |
| 32 | $\mathcal{D}_1$ | 0.127 | 25.91 | 0.0412 | 0.956 | |
| | $\mathcal{D}_2$ | 0.0885 | 25.42 | 0.0313 | 0.966 | $32 \to 64 \to 128$ |
| | $\mathcal{D}_3$ | 0.178 | 26.37 | 0.117 | 0.911 | |
| 64 | $\mathcal{D}_1$ | 0.0975 | 26.79 | 0.0086 | **0.965** | |
| | $\mathcal{D}_2$ | 0.0853 | 26.32 | 0.0084 | **0.974** | $64 \to 128$ |
| | $\mathcal{D}_3$ | 0.1907 | 27.23 | 0.0166 | **0.926** | |
| 128 | $\mathcal{D}_1$ | 0.0512 | 38.73 | - | 0.954 | |
| | $\mathcal{D}_2$ | 0.0453 | 34.16 | - | 0.968 | $128 \to 256$ |
| | $\mathcal{D}_3$ | 0.0726 | 43.42 | - | 0.917 | |
| 256 | $\mathcal{D}_1$ | 0.0543 | 41.20 | 0.0043 | 0.962 | |
| | $\mathcal{D}_2$ | 0.0301 | 38.96 | 0.0054 | 0.959 | $256 \to 512$ |
| | $\mathcal{D}_3$ | 0.0577 | 50.57 | 0.0103 | 0.904 | |

Table 14: **Number of codewords.** Experiment are on GTSRB dataset. "-" denotes value close to 0.

**Codebook Initialization.** Section 3.1 highlights our UEFL framework's capability for efficient codeword initialization via K-means, utilizing features from a trained encoder. The efficacy of K-means initialization is validated in Table 15 with results from the MNIST dataset, showing enhancements across all metrics.

**Extensible Codebook v.s. Static Large Codebook.** To validate our extensible codebook's superiority over starting with a large codebook, we ensured both methods ended with the same number of codewords through experiments. For the CIFAR100 dataset, the extensible codebook was initially set to 128 codewords and expanded twice, while the static codebook was fixed at 512 codewords. Results showcased in Table 16 demonstrate the difficulties associated with a larger initial codebook in codeword selection for image features. Conversely, gradually expanding the codebook significantly improved codeword differentiation, yielding better outcomes, such as enhanced accuracy (0.375 for Domain 1) and reduced uncertainty (0.78 vs. 1.66 for the static approach). In addition, perplexity results reveal increased utilization of our extensible codebook, offering clear evidence of our design's superiority.

**Different Uncertainty Threshold.** In our UEFL, the uncertainty evaluator plays a pivotal role in identifying heterogeneous data without needing direct data access, with the threshold selection being critical. An optimal threshold enhances the model's ability to distinguish between data silos, leading to quicker convergence. As illustrated in Figure 9, a lower threshold

Table 15: The experiments were conducted on the MNIST dataset with 128 initialized codewords and segmentation factor 1. The model with K-means initialization outperforms without it.

| Codebook | Data | $\mathcal{L}_{\mathbf{code}}$ | mP | mE | mA |
|---|---|---|---|---|---|
| w/o init | $\mathcal{D}_1$ | 0.6202 | 6.26 | 0.125 | 0.888 |
| | $\mathcal{D}_2$ | 0.6063 | 5.99 | 0.296 | 0.554 |
| | $\mathcal{D}_3$ | 0.6096 | 5.35 | 0.267 | 0.622 |
| **w/ init** | $\mathcal{D}_1$ | 0.0862 | 59.64 | 0.0935 | 0.945 |
| | $\mathcal{D}_2$ | 0.0785 | 57.58 | 0.1604 | 0.906 |
| | $\mathcal{D}_3$ | 0.0779 | 99.38 | 0.1509 | 0.929 |

Table 16: Our extensible codebook (Extend) outperforms the static larger codebook (Static) on all evaluation metrics.

| Codebook | Data | $\mathcal{L}_{\mathbf{code}}$ | mP | mE | mA |
|---|---|---|---|---|---|
| Static | $\mathcal{D}_1$ | 3.10 | 17.54 | 1.66 | 0.142 |
| | $\mathcal{D}_2$ | 2.91 | 17.41 | 1.76 | 0.135 |
| | $\mathcal{D}_3$ | 2.63 | 16.82 | 1.76 | 0.112 |
| **Extend** | $\mathcal{D}_1$ | 1.28 | 19.31 | 0.7822 | 0.375 |
| | $\mathcal{D}_2$ | 0.976 | 27.42 | 0.6665 | 0.341 |
| | $\mathcal{D}_3$ | 0.978 | 22.00 | 0.7112 | 0.304 |

imposes stricter criteria, pushing the model to achieve higher precision, thereby improving performance metrics. However, it's important to recognize that beyond a certain point, further reducing the threshold may not significantly enhance outcomes but will increase computational overhead. Thus, in such cases, there is a trade-off between runtime and performance.

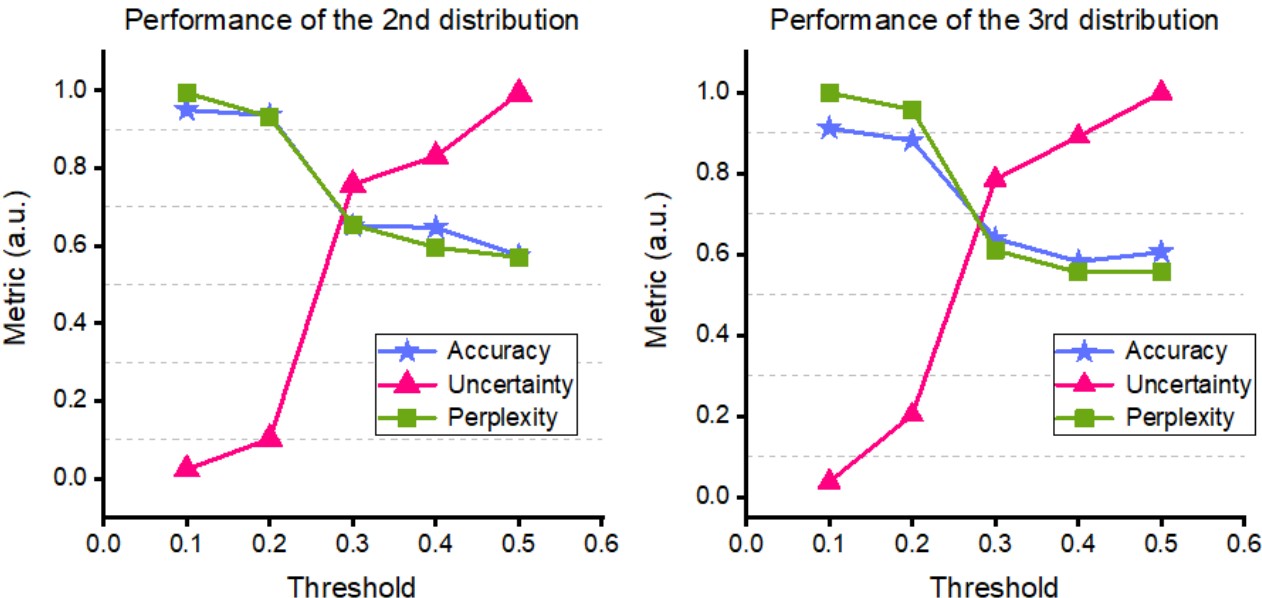

Figure 9: Results on GTSRB dataset with 64 initialized codewords with segment 1. Overall, a smaller threshold performs better.

