# OpenReview forum: "Uncertainty-Based Extensible Codebook for Discrete Federated Learning in Heterogeneous Data Silos"
_ICML.cc/2025/Conference — ICML 2025 poster_

### Official Review · Reviewer_SFNQ · 2025-03-13

**Overall Recommendation:** 3

**Summary:**

This paper introduces Uncertainty-Based Extensible-Codebook Federated Learning (UEFL), a novel framework designed to address data heterogeneity in federated learning (FL). The key innovation lies in dynamically mapping latent features to trainable discrete vectors (codewords) and extending the codebook for silos with high uncertainty, identified via Monte Carlo Dropout. UEFL demonstrates significant improvements in accuracy and uncertainty reduction across various datasets, including MNIST, CIFAR10, and CIFAR100. The extensible codebook approach, initialized using K-means, ensures efficient adaptation to unseen data distributions while maintaining low computational overhead.

**Claims And Evidence:**

The claims of improved accuracy and uncertainty reduction are well-supported by empirical evidence from experiments on multiple datasets.

**Essential References Not Discussed:**

No

**Experimental Designs Or Analyses:**

The experimental design is robust, with ablation studies and comparisons against strong baselines like FedAvg and DisTrans. The use of domain generalization tasks and large-scale setups (e.g., 50 and 100 clients) further strengthens the evaluation.

**Methods And Evaluation Criteria:**

The proposed method is well-suited to the problem of feature heterogeneity in FL, and the use of an extensible codebook with uncertainty evaluation is innovative. However, the introduction of an extensible codebook raises potential concerns regarding privacy risks that warrant further discussion. Since the codebook is shared and updated across clients, there may be a risk of leakage of sensitive information embedded in the codewords, especially if adversaries attempt to reverse-engineer the mapping between latent features and codewords.

While this paper emphasizes the alignment of codewords with latent features via K-means initialization to improve model performance, it remains unclear how this process is safeguarded against potential attacks, such as model inversion or membership inference attacks.

**Other Comments Or Suggestions:**

The resolution of the figures in the paper is quite low, and some of the text within the images is difficult to read due to its small size.

**Other Strengths And Weaknesses:**

No

**Questions For Authors:**

No

**Relation To Broader Scientific Literature:**

UEFL builds on prior work in FL, such as FedAvg and DisTrans, while introducing a novel mechanism to handle data heterogeneity through extensible codebooks and uncertainty-based adaptation. It also aligns with recent trends in uncertainty modeling (e.g., Monte Carlo Dropout, Deep Ensembles) and discrete representations for robustness.

**Theoretical Claims:**

This paper provides theoretical support for the benefits of discretization in reducing noise sensitivity and dimensionality, as outlined in Appendix A. While the provided theorems establish the foundational advantages, including enhanced robustness to noise and reduced dimensionality, a more detailed step-by-step proof process would further strengthen the theoretical claims and improve the clarity of the mathematical reasoning.

---

> ### Author Rebuttal · Authors · 2025-03-31
>
> We appreciate the recognition of the novelty of our method, and the robustness of our experimental design. Additionally, we value the insightful critique regarding the limitations of our work. In response, we address these issues below:
>
> > However, the introduction of an extensible codebook raises potential concerns regarding privacy risks that warrant further discussion. Since the codebook is shared and updated across clients, there may be a risk of leakage of sensitive information embedded in the codewords, especially if adversaries attempt to reverse-engineer the mapping between latent features and codewords.
>
> We thank you for highlighting the important concern regarding potential privacy risks associated with the extensible codebook. To ensure robust privacy preservation, our method follows standard federated learning protocols, raw data never leaves the local client, and only aggregated model updates are communicated. The codewords represent abstract latent features derived from model encoders, rather than explicit raw data or identifiable content. These latent vectors are highly compressed and abstract, significantly reducing the feasibility of reverse-engineering meaningful private information.
>
> Furthermore, the segmentation of codewords further abstracts feature information. Newly added codewords for highly heterogeneous silos are client-specific and accessible exclusively by those clients, minimizing potential information leakage risks. We will explicitly discuss these privacy aspects in our updated manuscript to highlight our framework's robust measures against potential privacy risks.
>
> Nonetheless, we acknowledge the insightful suggestion and completely agree that the introduction of an extensible codebook could raise potential privacy risks. However, this is not the focus of this work, as our primary aim was to address model accuracy and uncertainty reduction in heterogeneous data silos within federated learning settings. Possible solutions to address privacy concerns include differential privacy [1] or secure aggregation [2].
> We will leave the detailed exploration and integration of such privacy-preserving techniques for further study.
>
>
> > While this paper emphasizes the alignment of codewords with latent features via K-means initialization to improve model performance, it remains unclear how this process is safeguarded against potential attacks, such as model inversion or membership inference attacks.
>
> We appreciate this insightful comment regarding the robustness of our K-means initialization method against potential attacks. Our K-means algorithm operates solely on client-specific, highly abstracted latent embeddings rather than raw data, inherently limiting the risk of reconstructing original inputs. Additionally, our approach is inherently compatible with advanced privacy-preserving techniques, such as differential privacy [1], where calibrated noise can be added to the discrete codeword embeddings to further enhance security.
>
> Although our primary goal was to address model accuracy and uncertainty reduction, we fully acknowledge the concerns and will explicitly mention this security consideration in the revised manuscript, noting this as an area for further detailed exploration.
>
> > While the provided theorems establish the foundational advantages, including enhanced robustness to noise and reduced dimensionality, a more detailed step-by-step proof process would further strengthen the theoretical claims and improve the clarity of the mathematical reasoning.
>
> While Appendix A provides theoretical foundations supporting the benefits of discretization, we agree that providing a detailed, step-by-step derivation of our theoretical results would strengthen the clarity of our claims.
> In the revised supplementary materials, we will include full derivations based on Hoeffding Inequality for Theorem 1 and 2.
>
> > The resolution of the figures in the paper is quite low, and some of the text within the images is difficult to read due to its small size.
>
> Thank you for highlighting this issue. We have improved the readability and resolution of our figures by regenerating them using vector graphics in PDF format, with increased font sizes for better clarity. Please review a few updated figure samples (.pdf figures) at the following anonymous link: https://blush-melessa-85.tiiny.site
>
> Thanks once again for the constructive comments and valuable suggestions, which have significantly enhanced the quality and clarity of our manuscript.
>
> [1] Agarwal, Naman, Peter Kairouz, and Ziyu Liu. "The skellam mechanism for differentially private federated learning." Advances in Neural Information Processing Systems 34 (2021): 5052-5064.
>
> [2] Kairouz, Peter, Ziyu Liu, and Thomas Steinke. "The distributed discrete gaussian mechanism for federated learning with secure aggregation." International Conference on Machine Learning. PMLR, 2021.

---

### Official Review · Reviewer_t3oA · 2025-03-13

**Overall Recommendation:** 4

**Summary:**

The paper addresses the challenge of data heterogeneity in federated learning (FL) by proposing Uncertainty-Based Extensible-Codebook Federated Learning (UEFL). The method dynamically extends a codebook of latent vectors using uncertainty estimates (via Monte Carlo Dropout) to adapt to diverse data distributions across silos. Key innovations include K-means initialization for new codewords, segmentation of feature vectors, and iterative codebook expansion. Experiments on rotated datasets (MNIST, CIFAR, etc.) demonstrate improvements in accuracy (3%-22.1%) and uncertainty reduction (38.83%-96.24%) over FedAvg and DisTrans. The approach also scales well to large client numbers (50–100) and handles domain generalization tasks.

**Claims And Evidence:**

The claims in the paper are supported by clear evidence:
1. The proposed method shows consistent improvement in MNIST, PACS, CIFAR10, and CIFAR 100. The authors have designed sufficient experiments to prove the effectiveness of the proposed UEFL.
2. The authors have conducted careful ablation studies on the hyperparameters (e.g. Uncertainty Threshold)
3. The paper also gives a theoretical explanation in Appendix A.

**Essential References Not Discussed:**

The authors discussed the related works well.

**Experimental Designs Or Analyses:**

The paper have conducted a series of experiments on MNIST, PACS, CIFAR10, and CIFAR 100. Careful ablation studies have been done.

**Methods And Evaluation Criteria:**

Yes, the proposed method is simple and straightforward in solving the problem of limited codebooks when heterogeneous data occurs. It is expected to get an improvement. The experiments also prove the method's effectiveness.

**Other Comments Or Suggestions:**

NA

**Other Strengths And Weaknesses:**

Strength:
1. The algorithm introduces light overhead, which is suitable for edge deployment.
2. The paper is well written and easy to read.

**Questions For Authors:**

1. What is the extra information, except for the model parameters, that the clients send to the central server? Is there any risk of information leakage during the codebook sharing?

**Relation To Broader Scientific Literature:**

The paper is related to privacy-preserving training.

**Theoretical Claims:**

There is no proof in the paper.

---

> ### Author Rebuttal · Authors · 2025-03-31
>
> We sincerely thank you for the insightful and constructive feedback, as well as the recognition of our contributions and experiments. Below, we address the specific questions raised:
>
> > What is the extra information, except for the model parameters, that the clients send to the central server? Is there any risk of information leakage during the codebook sharing?
>
> In our UEFL framework, the codebook is a trainable component of the model architecture (like classifier weights or encoder layers) and is thus included in the standard model parameters exchanged during federated averaging. Clients only share updated model parameters (including discrete codebook vectors) with the server, no raw data or additional metadata.
>
> Codebooks map latent features to discrete codewords, which represent aggregated and abstracted latent features rather than raw data, less prone to information leakage. Discrete representations inherently limit the granularity of shared information, aligning with privacy-preserving mechanisms in federated learning [1, 2]. In addition, the segmentation of codewords (Section 3.2) further abstracts feature information, enhancing the robustness against potential information leakage compared to raw data or explicit feature representations.
>
> We will clearly clarify these points in the revised manuscript to explicitly address potential privacy concerns. We deeply appreciate the thoughtful review and suggestions.
>
> [1] Kairouz, Peter, Ziyu Liu, and Thomas Steinke. "The distributed discrete gaussian mechanism for federated learning with secure aggregation." International Conference on Machine Learning. PMLR, 2021.
>
> [2] Agarwal, Naman, Peter Kairouz, and Ziyu Liu. "The skellam mechanism for differentially private federated learning." Advances in Neural Information Processing Systems 34 (2021): 5052-5064.

---

> > ### Comment · Reviewer_t3oA · 2025-04-08
> >
> > Thank you for the detailed explanation. The answer already delivers my problem.

---

> > > ### Author Response · Authors · 2025-04-08
> > >
> > > We greatly appreciate your positive feedback and acknowledgment. Thank you again for your insightful comments and helpful suggestions.

---

### Official Review · Reviewer_idfz · 2025-03-13

**Overall Recommendation:** 3

**Summary:**

The paper introduces Uncertainty-Based Extensible-Codebook Federated Learning to address data heterogeneity in federated learning (FL) by dynamically expanding a discrete codebook based on model uncertainty. UEFL improves generalization by mapping latent features to trainable codewords and selectively extending the codebook for clients exhibiting high uncertainty, reducing performance degradation caused by non-IID data distributions. The approach integrates Monte Carlo Dropout for uncertainty evaluation and K-means clustering for efficient codeword initialization, ensuring minimal computational overhead.

**Claims And Evidence:**

The computational overhead of UEFL is minimal and does not impact scalability. While UEFL’s memory overhead is shown to be small (~3.34% increase), scalability to thousands of clients is not tested. It is also useful to provide the estimation of the overhead w.r.t. the number of clients.

**Essential References Not Discussed:**

NA

**Experimental Designs Or Analyses:**

Figure 9 shows that lower thresholds improve performance, but there is no principled way to set the threshold.

**Methods And Evaluation Criteria:**

FL often deals with privacy-preserving scenarios where clients cannot share model updates freely. The paper assumes all clients can exchange information, but in some FL settings, stricter constraints (e.g., differential privacy, homomorphic encryption) exist.

**Other Comments Or Suggestions:**

1. The plots are blurry when I zoom in. It is better to use vector illustration.
2. The running title needs an update.
3. The experimental setting is unclear. It says the "experiments are performed on a machine with 2 GPUs". Did you use both GPUs or only one GPU?

**Other Strengths And Weaknesses:**

The paper does not analyze long-term codebook growth, which could become a computational bottleneck over extended training.

This work lacks theoretical support, e.g., analysis of the size of the codebook on the uncertainty.

**Questions For Authors:**

1. How does performance degrade if the codebook is not expanded enough?

2. The baseline comparison primarily includes FedAvg (2017) and DisTrans (2022), which are outdated given recent advancements in federated personalization and adaptive aggregation methods; incorporating newer techniques like FedPer, FedRod, FedBABU, or FedDyn would provide a more rigorous evaluation of UEFL’s effectiveness.

3. This work utilizes a pre-trained VGG model for tasks like CIFAR and GTSRB, which already have strong feature extractors. How does this justify the necessity of the proposed approach, given that the model is already well-suited for these datasets?

**Relation To Broader Scientific Literature:**

NA

**Theoretical Claims:**

The theoretical section (Appendix A) provides proofs justifying discretization in FL. However, they rely on IID assumptions, while real FL data is often non-IID.

---

> ### Author Rebuttal · Authors · 2025-03-31
>
> We appreciate the insightful comments and suggestions. In response, we address these issues below:
>
> > Scalability to thousands of clients is not tested. Estimate overhead w.r.t. the number of clients
>
> While we did not test UEFL with thousands of clients, we evaluated it with 50 & 100 clients, showing consistent improvements over baselines (Tables 3 & 4). UEFL’s overhead scales with iteration count, not the number of clients. In each iteration, 64 codewords are added and shared with all selected high-uncertainty clients; the overhead after $i$ iterations is:
>
> \begin{equation}
>     Overhead(i) = \frac{i\times 0.125}{14.991}\times 100\\%
> \end{equation}
>
> UEFL typically needs 1-3 iterations and we set the maximum to 5 (Section 3.2), so the introduced overhead is at most 4.17%.
>
> > In some FL settings, stricter constraints (e.g., differential privacy, homomorphic encryption) exist
>
> Thanks for the comment. Our current experimental setup assumes standard FL settings, but UEFL is compatible with these privacy-preserving methods. Specifically, our codeword-based discretization process can integrate differential privacy by adding calibrated noise to codeword embeddings. We will explore this in future work.
>
> > The theoretical section ... rely on IID assumptions, while real FL data is often non-IID
>
> In non-IID FL with $K$ clients, client $k$ has $n_k$ samples drawn from its distribution $P_k$. Total samples $n = \sum_{k=1}^Kn_k$. The global distribution is $\overline{P} = \sum_{k=1}^K \frac{n_k}{n}P_k$, then
>
> **With discretization:**
>
> \begin{equation}
>     \left|\sum_{k=1}^K \frac{n_k}{n} \mathbb{E}_{\boldsymbol{h} \sim P_k}[\phi_k^S(q(\boldsymbol{h}, L, G))] - \frac{1}{n} \sum\_{k=1}^K \sum\_{i=1}^{n_k} \phi_k^S(q(\boldsymbol{h}_i^{(k)}, L, G))\right| = \mathcal{O}( \alpha \sqrt{ \frac{G \ln L + \ln(2K/\delta)}{2n}} + \frac{\nu^{(q)}}{\sqrt{n}}),
> \end{equation}
>
> **Without discretization:**
>
> \begin{equation}
>     \left|\sum_{k=1}^K\frac{n_k}{n} \mathbb{E}_{\boldsymbol{h} \sim P_k}[\phi_k^S(\boldsymbol{h})] - \frac{1}{n} \sum\_{k=1}^K \sum\_{i=1}^{n_k} \phi_k^S(\boldsymbol{h}_i^{(k)}) \right| = \mathcal{O}( \alpha \sqrt{ \frac{m \ln(4\sqrt{nm}) + \ln(2K/\delta)}{2n} } + \frac{\overline{\varsigma} R\_\mathcal{H} + \nu}{\sqrt{n}}),
> \end{equation}
>
> Here, $\nu^{(q)} = \frac{1}{K}\sum_{k=1}^K\text{Div}(P_k^{(q)}, \overline{P}^{(q)})$ and $\nu = \frac{1}{K}\sum_{k=1}^K\text{Div}(P_k, \overline{P})$ denote the KL divergence between client and global distribution with and without discretization. $\nu^{(q)} < \nu$.
> Therefore, discretization not only improves robustness to noise and reduces dimensionality, but also it effectively mitigates the effects of data heterogeneity typical in non-IID FL.
>
> > Figure 9 ... no principled way to set threshold
>
> In this work, the threshold is manually tuned per dataset. We agree that a dynamic adjustment mechanism (e.g., based on convergence) is promising future work.
>
> > Not analyze long-term codebook growth ... analysis of the size of the codebook on the uncertainty
>
> Computation is discussed above.
>
> Codeword utilization, measured by perplexity ($\exp(-\sum_{\text{class}} p \log p)$), initially increases as the codebook expands, leading to richer representations and higher mutual information $I(Z; Y)$, which reduces uncertainty ($H(Y|Z) = H(Y) - I(Z; Y)$). However, when the codebook becomes very large, most codewords are rarely used (collapse) [1], resulting in reduced perplexity and diminishing returns in uncertainty reduction. More details will be included.
>
> > The plots are blurry. Use vector illustration
>
> Thanks. We have replaced all figures with vector graphics to ensure clarity. Please review a few updated samples (.pdf figures): https://blush-melessa-85.tiiny.site
>
> > The running title needs an update.
>
> We have updated it to: "UEFL: Uncertainty-Based Extensible Codebook Federated Learning".
>
> > The experimental setting is unclear ... Use both GPUs or only one?
>
> All experiments utilized only one GPU, added to the revision.
>
> > How does performance degrade if the codebook is not expanded enough?
>
> Then, the model lacks representational capacity, resulting in reduced accuracy and higher uncertainty (Figure 6). However, UEFL typically converged after 1-3 iterations.
>
> > Incorporate newer techniques like FedPer, FedRod, FedBABU, or FedDyn
>
> Here is the comparison with FedRod & FedDyn:
> | Method| FMNIST|
> | -------- | ------- |
> | FedDyn| 89.65   |
> | FedRod| 90.28   |
> | UEFL| 90.59    |
>
> UEFL outperforms them. More results will be included in the updated version.
>
> > This work utilizes a pre-trained VGG model ... justify the necessity of the proposed approach, given that the model is already well-suited for these datasets?
>
> While the encoder is strong, the classifier is still vulnerable to domain shifts. UEFL improves performance via discretization, as shown in Table 1.
>
> [1] Huh et al., Straightening out the straight-through estimator: Overcoming optimization challenges in vector quantized networks, ICML 2023.

---

### Official Review · Reviewer_7WL8 · 2025-03-16

**Overall Recommendation:** 3

**Summary:**

This paper introduces Uncertainty-Based Extensible-Codebook Federated Learning (UEFL), a novel framework addressing data heterogeneity in federated learning. The key idea is to dynamically extend a codebook of discrete latent vectors based on model uncertainty, which is evaluated via Monte Carlo Dropout. UEFL initializes a small shared codebook and iteratively adds client-specific codewords using K-means clustering on encoder features for underrepresented distributions. Experiments on rotated MNIST, CIFAR, GTSRB, and PACS datasets demonstrate improvements in accuracy and uncertainty reduction compared to FedAvg and DisTrans. The method also shows scalability to large client numbers and robustness in domain generalization tasks.

**Claims And Evidence:**

Yes

**Essential References Not Discussed:**

VQ-FL (Chen et al., 2023, ICML): Uses vector quantization for client-specific representation learning but fixes the codebook size. UEFL’s uncertainty-driven extension is novel, but a comparison is necessary.

FedPM (Dinh et al., 2022, NeurIPS): Personalizes models via latent mask vectors. While distinct from codebooks, its focus on client-specific latent spaces is conceptually related.

FedProx (Li et al., 2020, MLSys): Addresses heterogeneity via proximal regularization. Although cited in the related work, its comparison with UEFL in terms of uncertainty reduction is missing.

**Experimental Designs Or Analyses:**

Limited simulation of data heterogeneity: The paper mainly uses rotation transformation to introduce heterogeneity, but in real federated learning environments, data heterogeneity usually includes uneven class distribution, feature space shift, etc., and does not cover more complex distribution drift situations.

Lack of generalization analysis for different codebook sizes: Although the paper provides some experiments with different codebook sizes
such as K-means initialization vs. random initialization), there is a lack of detailed discussion on the impact of different codebook sizes on model stability.

Insufficient ablation experiments for uncertainty assessment methods: Although the paper compared Deep Ensemble, the experiment only used 5 sub models and did not explore whether increasing the number of sub models would affect the stability of the assessment.

**Methods And Evaluation Criteria:**

Yes

**Other Comments Or Suggestions:**

See the weakness

**Other Strengths And Weaknesses:**

Strengths:

A dynamic codebook extension strategy based on uncertainty assessment has been proposed, which is more adaptable compared to existing discretization FL methods such as VQ FedAvg.

Using K-means for codebook initialization reduces the instability caused by random initialization and improves model training efficiency.

Multiple datasets (MNIST, FMNIST, CIFAR10, CIFAR100, GTSRB) were used for experiments, and rotation transformation was employed to simulate data heterogeneity.

Weaknesses:

The paper provides a theoretical analysis of the discretization of generalization error in Appendix A, but this analysis relies on the i.i.d. assumption and does not consider the Non IID distribution in FL.

The paper does not provide a mathematical analysis of the effect of codebook size on convergence, and only verifies its impact through experiments.

The paper mainly simulates Feature Heterogeneity, but there is limited exploration of Label Heterogeneity, with only limited experiments conducted in Appendix H.

The paper only uses K-means for initialization of the codebook and does not explore other possible initialization methods (such as PCA dimensionality reduction and contrastive learning feature initialization).

**Questions For Authors:**

Communication Overhead: How does UEFL’s communication cost (e.g., transmitting codeword vectors) scale with the number of clients and codebook size? For 100 clients (Table 4), does the server need to aggregate 100 unique codebooks?

Theoretical Grounding: Can Theorem 1 be extended to account for dynamic codebook growth? For example, does adding codewords tighten the generalization bound in Eq. 8?

The mathematical analysis of the paper assumes that the data is i.i.d., but FL is usually a Non IID scenario. Is there an experiment conducted under more extreme Non IID settings (such as Dirichlet distribution α=0.01)? Is the scalability of UEFL still effective for situations where data distribution is severely uneven across different clients?

**Relation To Broader Scientific Literature:**

The core contributions of this paper mainly involve two fields: Federated Learning (FL) and Uncertainty Modeling, and are related to existing research as follows:

Data heterogeneity is a key challenge in FL, and various methods have been proposed in previous studies to alleviate this issue: Personalized FL based methods (such as FedPer, FedRep): Processing heterogeneous data through hierarchical separation or personalized local models [Li et al., 2021]. FL methods based on distribution transformation (such as DisTrans): Processing data heterogeneity through distribution transformation during training and testing [Yuan et al., 2022]. FL methods based on knowledge distillation (such as FCCL): using knowledge distillation and unlabeled public data to enhance generalization ability [Huang et al., 2022]. The core innovation of UEFL in this article lies in the extensible codebook, which is similar to the idea of enhancing FL robustness through discretization (such as VQ FedAvg [Liu et al., 2021]). However, this article additionally introduces a dynamic codebook extension based on uncertainty to adapt to data with larger distribution biases.

Previous studies have used uncertainty quantification to improve the robustness of FL models: Method based on Monte Carlo Dropout (Gal&Ghahramani, 2016): Estimating uncertainty by enabling Dropout during the inference phase for data selection and model weighting. Method based on Deep Ensemble (Lakshminarayanan et al., 2017): Train multiple models and estimate uncertainty through analysis of variance. This article uses Monte Carlo Dropout as an uncertainty assessment method and combines it with K-means to dynamically extend the codebook, further expanding the application of uncertainty modeling in the field of FL.

The paper mentions VQ-VAE [Van Den Oord et al., 2017] as inspiration for using a discrete codebook for feature mapping. In the field of FL, Liu et al.,  The use of discretization to enhance model generalization ability has been proposed in 2021, but the encoding of this method is fixed. This article proposes an extensible codebook mechanism and dynamically adjusts the codebook size by combining uncertainty, which is one of the main innovations of this article.

In summary, the innovation of this paper mainly lies in the combination of uncertainty quantification and scalable codebook mechanism to handle FL data heterogeneity, and empirical research on multiple datasets. These contributions complement existing literature, but more extensive experiments are still needed to verify their applicability (such as other data types, communication overhead, etc.).

**Theoretical Claims:**

Theorems 1 and 2 in Appendix A argue that discretization reduces generalization gaps by lowering noise sensitivity and dimensionality. While the theorems are logically structured, their connection to UEFL’s empirical success is not explicitly discussed. For instance, how the codebook’s extensibility interacts with the theoretical guarantees remains unclear. The proof relies on Hoeffding's Inequality, but does not explicitly discuss which distribution data the inequality applies to, especially in the Non IID scenario of federated learning, which may affect its applicability.

---

> ### Author Rebuttal · Authors · 2025-03-31
>
> Thanks for the insightful comments and suggestions. We address your concerns as follows:
>
> > While the theorems ... connection to UEFL’s empirical success is not explicitly discussed
>
> We provide a detailed analysis in response to the later theoretical questions. Please refer to that.
>
> > The paper mainly uses rotation transformation ... not cover more complex distribution drift situations
>
> Other forms of heterogeneity are also included:
> - Feature space shifts: Tables 2 and 3 (e.g., PACS domains)
> - Uneven class distributions: Table 12 (Label heterogeneity)
>
> > Lack of detailed discussion on the impact of different codebook sizes on model stability
>
> We ran experiments across 5 random seeds. As the codebook size increases from 16 to 64, the standard deviation of accuracy drops from 0.0237 to 0.0094 (higher stability).
>
> > Only used 5 sub models ... not explore whether increasing the number of sub models would affect the stability of the assessment
>
> We extended the ensemble size to 20 and observed the following on MNIST:
> | Method | Accuracy| Uncertainty|
> | ----- | ----- |----- |
> | FedAvg | 0.782 | 0.261|
> | UEFL | 0.924 | 0.237|
> UEFL also outperforms. However, consistent with prior findings (e.g., Lakshminarayanan et al., 2017), we observed that minimal improvement beyond 5–10 ensembles while computation costs increase linearly.
>
> > Appendix B ... not provide a detailed generalization error curve
>
> We have plotted the curve: https://ibb.co/xymvPDT
>
> > Appendix F ... does not explore the impact of communication costs
>
> In practice, UEFL uses 30 communication rounds in its first iteration, compared to 40 in standard FL. Each subsequent iteration requires only 5 rounds with a modest model size increase of 0.83%. For example, with two additional iterations, the overall communication cost is approximately $\frac{30}{40}+\frac{5}{40}\times1.0083+\frac{5}{40}\times1.0167 = 1.0031\times$ of standard FL.
>
> > More extensive experiments are still needed to verify their applicability
>
> We’re extending this to medical data and other domains in future work.
>
> > Essential References Not Discussed
>
> Thanks. Specifically for FedProx, the results are as follows:
> | Method| FMNIST|
> | -------- | ------- |
> | FedProx| 88.55   |
> | UEFL| 90.59    |
> we will add more comparisons in the updated version.
>
> > The paper provides a theoretical analysis ... not consider the Non IID distribution in FL
>
> In non-IID FL, then
>
> **With discretization:**
> \begin{equation}
>    \mathcal{O}( \alpha \sqrt{ \frac{G \ln L + \ln(2K/\delta)}{2n}} + \frac{\nu^{(q)}}{\sqrt{n}}),
> \end{equation}
> **Without discretization:**
> \begin{equation}
>     \mathcal{O}( \alpha \sqrt{ \frac{m \ln(4\sqrt{nm}) + \ln(2K/\delta)}{2n} } + \frac{\overline{\varsigma} R\_\mathcal{H} + \nu}{\sqrt{n}}),
> \end{equation}
> Here, the KL divergence $\nu^{(q)} < \nu$.
> Therefore, discretization explicitly mitigates the effects of data heterogeneity.
>
> To conserve space, more details are included in our response to Reviewer idfz.
>
> > Not provide a mathematical analysis of the effect of codebook size on convergence
>
> The basic mathematical analysis follows VQ-VAE. Large codebooks hinder convergence due to poor utilization and noisy updates. To address this, UEFL progressively extends the codebook, initializing new codewords with K-means for better alignment.
> This improves codeword utilization, reduces quantization error, and accelerates convergence (Fig. 3(b)).
>
> > Limited exploration of Label Heterogeneity
>
> We further test different dirichlet distributions as follows,
> | Method| α=0.05| α=0.01|
> | ----- | ----- |----- |
> | FedAvg| 78.57   |35.42   |
> | UEFL|85.18    | 40.99   |
> UEFL performs better. More results will be included.
>
> > Only uses K-means for initialization ... not explore other possible initialization methods
>
> Thanks. We agree that PCA or contrastive-based initialization is promising and will explore them in future work.
>
> > How does UEFL’s communication cost (e.g.,transmitting codeword vectors) scale with the number of clients and codebook size? For 100 clients (Table 4), does the server need to aggregate 100 unique codebooks?
>
> No. Codebook growth is tied to iterations, not client count. In each iteration, 64 codewords are added and shared with selected high-uncertainty clients. The communication overhead after $i$ iterations is $Overhead(i) = \frac{i\times 0.125}{14.991}$.
> With a typical $i=1-3$ and $i=5$ (Section 3.2), the worst-case overhead is under 4.17% (5 unique codebooks), regardless of client count (100).
>
> > Can Theorem 1 be extended to account for dynamic codebook growth?
>
> Yes. Although Theorem 1 assumes static $L$, dynamic growth adds a minor $\text{ln}L$ term. $L$ remains small (e.g., 64–256). However, dynamic codebook growth improves representation capacity, thereby reducing the KL divergence term $v^{(q)}$, tightening the federated generalization bound.
>
> > Experiment conducted under more extreme Non IID settings (α=0.01)?
>
> The results for Dirichlet distribution α=0.01 are discussed above.

---

### Decision · Program_Chairs · 2025-05-01

**Decision:**

Accept (poster)

**Comment:**

This paper has addressed the data distribution shift in discrete federated learning. The core idea is to dynamically extend a codebook of latent vectors using uncertainty estimates, i.e., Monte Carlo Dropout, to adapt to diverse data distributions across silos. The problem is well defined, and the method has been well formulated. Some good results have been made.

Some suggestions, i.e.,

- Include the experiments added in rebuttal in the final version.
- Revise some figures with low resolution.